# Phylogeochemistry: exploring evolutionary constraints on belemnite rostrum element composition

Alexander Pohle[1], Kevin Stevens[1], René Hoffmann[1], and Adrian Immenhauser[1,2]

[1]Institute of Geology, Mineralogy and Geophysics, Ruhr University Bochum, Universitätsstraße 150, 44799 Bochum, Germany

[2]Fraunhofer IEG (Fraunhofer Institution for Energy Infrastructures and Geothermal Systems), Am Hochschulcampus 1, 44801 Bochum, Germany

**Correspondence:** Alexander Pohle (alexander.pohle@rub.de)

**Abstract.** The biogenic carbonate hardparts of a large range of marine organisms are among the most important geochemical archives of Earth's climate dynamics through time and the evolution of life. That said, biomineralization pathways, i.e., the secretion of mineral phases by organisms, are complex and may differ significantly between different taxa. In light of this, it is critically important to evaluate if related taxa might display similar hard parts geochemistry. If so, this relation might

bear information on evolutionary relationships and has great significance in carbonate archive research. Here, we test the evolutionary constraints on element/Ca ratios of belemnite rostra using Bayesian phylogenetic tools. For this purpose, we assembled a large dataset on element ratios from 2241 published samples of belemnite rostra. We used comparative Bayesian phylogenetic tools to reconstruct ancestral states and evolutionary rates of these geochemical data based on trees inferred from morphological data. While Mn/Ca and Fe/Ca appear to be taxon-independent and probably mainly reflect environmental and

diagenetic effects, Mg/Ca and Sr/Ca display stronger taxon-specific patterns, even though their interpretation remains complex. The evolutionary rates are high, with average estimated changes in element ratios of 12.4% (Mg/Ca; confidence interval 9.0-16.9%) and 12.3% (Sr/Ca; confidence interval 5.5-18.3%) of the overall mean element ratio within 1 million years (Myr). While the distribution of Sr/Ca ratios is relatively homogeneous across the tree, Mg concentrations are divided among two distinct groups (< 5.5 and >7.5 mmol/mol, respectively), with at least five evolutionary transitions between them. Beyond

carbonate archive research, our phylogenetic analysis provides insights into the evolution of belemnites. This study highlights the complex interplay between evolutionary, ontogenetic, environmental and diagenetic effects and calls for caution when using belemnite element ratios as proxies for palaeoclimatic studies. We propose the term 'phylogeochemistry' for the investigation of geochemical data using phylogenetic modelling techniques.

## 1   Introduction

Belemnites are a group of coleoid cephalopods, likely representing stem group decabrachians (today's squids, cuttlefishes and relatives) that were particularly abundant and diverse in mainly open marine waters of the Jurassic and Cretaceous oceans (Hoffmann and Stevens, 2020 and references therein). Their calcitic internal skeletal element, the rostrum, is a common fossil in Mesozoic marine sedimentary rocks and a widely used carbonate archive (e.g., Urey, 1948; Urey et al., 1951; Lowenstam and

Epstein, 1954; Spaeth, 1971; Rosales et al., 2004; Dutton et al., 2007; Arabas, 2016). Palaeoenvironmental parameters based on belemnite archive calcite include seawater oxygenation level, seawater temperature, alkalinity (pH) or seawater isotope and elemental data, among others (Wierzbowski, 2004; Gómez et al., 2008). Further, the calcite of a belemnite rostrum from the Upper Cretaceous Peedee Formation also served as the base for the PeeDee Belemnite standard (now Vienna PeeDee Belemnite, VPDB; Slater et al., 2001) in analytical geochemistry. Despite their wide distribution in (hemi)pelagic sections worldwide, and regardless of many decades of dedicated research, the interpretation of belemnite geochemical data is still fraught with complications.

Specifically, complexity results from (i) metabolic processes, i.e., biomineralization strategies, likely resulting in ontogenetic trends in sclerochronologically sampled rostra (Spaeth, 1971; Vonhof et al., 2011; Ullmann et al., 2015; Immenhauser et al., 2016). (ii) Further, the complex rostrum calcite ultrastructure is not well understood. High-resolution petrographic studies of rostra from multiple belemnite genera revealed the presence of two distinct calcite phases (Benito et al., 2016; Hoffmann et al., 2016, 2021). The first phase represents an (arguably porous) biomineral fabric secreted during the lifetime of the animal from extrapallial fluids (geochemically close to the ambient seawater), while the second phase is less well constrained in terms of its formation time and arguably had an amorphous precursor phase (Hoffmann et al., 2021). (iii) Post-mortem diagenetic alteration processes commonly complicate these patterns. Specifically, the organic-rich layers in the rostrum were subject to microbial biodegradation, with microbes targeting the organic matter. The relation between organic matter, biodegradation and diagenetic alteration is a typical feature of all marine skeletal carbonates (Lange et al., 2018). During the subsequent burial, alteration is most pronounced and can cause the recrystallisation of portions (or all) of the rostrum (Wierzbowski and Joachimski, 2009; Immenhauser et al., 2016). Numerous geochemical and optical screening methods have been proposed to separate well-preserved from altered material (e.g., Veizer, 1974, 1983; Saelen, 1989; Ullmann and Korte, 2015; Stevens et al., 2017). (iv) Moreover, significant variation in geochemical proxies, such as element concentrations or isotope data between different belemnite species sampled at the same stratigraphical level, was observed (Spaeth et al., 1971; Niebuhr and Joachimski, 2002; McArthur et al., 2004, 2007; Mutterlose et al., 2010; Li et al., 2013; Stevens et al., 2022). This is particularly noteworthy as trace element concentrations have been shown to be phylogenetically correlated in living calcifying organisms (Ulrich et al., 2021), and taxon-specific signals in element/Ca ratios have been shown in brachiopods (e.g., Grossman et al., 1996; Ullmann et al., 2017). In a similar context, recent studies have documented that biomineralization patterns are strongly constrained by phylogeny. Examples include the calcite-aragonite ratios in modern bryozoans (Piwoni-Piórewicz et al., 2024) or the crystal orientation in conodont dental elements (Shirley et al., 2024). However, a similar phylogenetic perspective on element incorporation into belemnite rostra has not been studied systematically.

This omission forms a strong motivation for the present paper proposing the term 'phylogeochemistry' for the rigorous combination of phylogenetic tools and geochemical proxy data in rostra. The working hypothesis is that the rostra of different clades of belemnites differ in their geochemical properties. If that axiom holds, and acknowledging the intrinsic complexity of belemnite geochemistry in general, then these data have the potential to be used as a character in elucidating phylogenetic relationships of belemnites, which are currently challenging to constrain due to the scarcity of morphological characters. First quantitative efforts in this direction have only been explored recently (Stevens et al., 2023). Beyond phylogenetic relation-

ships, understanding taxonomic patterns in belemnite rostrum geochemistry has wide significance for studies with a focus on

palaeoceanography.

This paper has the following aims: (i) A large (literature) dataset of taxon-specific element/Ca ratios and the application of ancestral state estimation approaches serves to test whether these proxy data are phylogenetically constrained. Importantly, this means that we do not use geochemical data to infer phylogenetic relationships, but rather compare how they change across a tree reconstructed by morphological data. (ii) Based on this, we explore how to interpret geochemical data from an evolutionary

perspective. For the sake of focus, we limit this study to a set of element/Ca ratios (Mg/Ca, Sr/Ca, Fe/Ca and Mn/Ca). The application of phylogeochemistry to isotope data from belemnite rostra must be the focus of forthcoming work.

Our article is aimed at researchers with different scientific backgrounds, such as evolutionary biologists and palaeontologists on the one hand and geochemists and oceanographers on the other hand. This might imply that part of the terminology will be unfamiliar to potential readers. To improve the accessibility of the science presented here, we provide definitions of important

terms in Table 1. For further reading, we recommend scholarly overview papers of the respective literature for Bayesian phylogenetics (e.g., Wright, 2019; Wright et al., 2022), belemnite palaeobiology (Hoffmann and Stevens, 2020) and geochemical proxy data and their interpretation (e.g., Veizer, 1974; Rosales et al., 2004; Schöne, 2008; Swart, 2015; Ullmann et al., 2015; Immenhauser et al., 2016; Immenhauser, 2022, and references therein).

## 2  Methodology

An overview of the methodology applied is given in Fig. 1. The approach used is divided into three main work packages: (i) acquisition of geochemical data from the literature (Fig. 1f-i), (ii) Bayesian inference of an updated belemnite phylogeny to include taxa for which geochemical data is available, based on morphological characters (Fig. 1a-e) and (iii) evolutionary rate estimation and ancestral state reconstructions (Fig. 1j-l) based on trees from (i) and data from (ii). Details on each of these work packages are given below.

### 2.1  Element/Ca ratio data

We compiled a large belemnite rostrum dataset of element concentrations (expressed as element/Ca ratios) from the published literature (Table 2, Fig. 1f). Specifically, we compiled Mg/Ca, Sr/Ca, Fe/Ca and Mn/Ca ratios, in addition to the elemental concentrations for Ca (in wt%), Mg, Sr, Fe and Mn (usually in ppm = $\mu g/g$), where available. We only included data from studies that identified rostra at least to the genus level. We excluded data when the authors found evidence for incipient to

pervasive diagenetic alteration. That said, we cannot rule out that some of the data from rostra - that were considered as well-preserved by the authors - yielded variable (undetected) alteration features and, hence, reset elemental concentrations. After some consideration, this potential shortcoming was accepted because we aimed to (i) maximise the dataset, (ii) cover a large range in sampling localities across the world and from different stratigraphic levels, (iii) include several genera recorded from multiple studies and collection sites. Furthermore, to avoid circular reasoning and to keep the values comparable between

studies, we decided to still include some data points where alteration had been suspected based on specific threshold values

**Table 1.** Definition of important terms and concepts used in this paper. See Felsenstein (2003) for a comprehensive introduction to phylogenetic methods. Further references are listed below (see also references therein).

| Term | Explanation |
| --- | --- |
| Tip | The endpoints of a phylogenetic tree, usually representing species or other taxonomic units (operational taxonomic unit = OTU). |
| Internal node | The points of divergence within a phylogenetic tree. Each node represents a bifurcation with exactly one parent node (internal node or sampled ancestor) and two child nodes (tip, internal node or sampled ancestor). |
| Branch | The connections between tips, internal nodes and sampled ancestors, essentially representing unsampled lineages. |
| Tree topology | The shape of a phylogenetic tree, i.e., how tips, internal nodes and sampled ancestors are connected, irrespective of branch lengths. |
| Posterior sample | The outcome of a Bayesian analysis, containing a distribution of values (or trees) for each parameter of interest that reflect uncertainty in the results. (Wright, 2019) |
| Ancestral state reconstruction | A set of techniques that aim to reconstruct the ancestral condition (= state) of a trait (= character) at internal nodes of a (usually fixed) tree. The conditions may be known in tips and sampled ancestors, from which the condition in all nodes and OTUs with unknown states is inferred. It is also possible to simulate changes along branches. (Joy et al., 2016) |
| Evolutionary rate (Brownian motion) | In the context of this paper, evolutionary rate refers to $\sigma^2$, which is the rate of change in a continuous character (here: element/Ca ratios). Specifically, the character evolves on a branch of length $t$ with an average change of 0.0 and a variance of $t \times \sigma^2$. Thus, the standard deviation of the character (i.e., the average absolute change) within 1 Myr is $\sigma$, and $\sqrt{(10 \times \sigma^2)}$ within 10 Myr. |
| Diagenesis | All chemical, mineralogical, physical, and (micro)biological changes underwent by a sediment after its initial deposition or biological secretion and during and after its lithification, exclusive of weathering and preceding very low-grade metamorphism (Immenhauser, 2022). |
| Non-equilibrium (kinetic) carbonate precipitation | Precipitation of carbonate (bio)minerals from a fluid, including bodily fluids that are reproducibly out of isotopic equilibrium with the environmental solution (Arvidson and Mackenzie, 1999). |
| Biomineralization | The process in which living organisms produce minerals or induce mineralization via their metabolic products. Organisms have been producing mineralized skeletons for the past 570 Myr (Cuif et al., 2010). |
| Carbonate archive | Biogenic and abiogenic carbonate phases and their proxy data represent snap-shots of the evolution of Earth's surface palaeoenvironments. A fossil carbonate archive can potentially be dated and serves as a source of proxy data (geochemical or petrographic evidence). By means of proxy data, a desired but no longer accessible environmental parameter (such as seawater temperature and the like; Mueller et al., 2024) can be, under favourable conditions, reconstructed. |

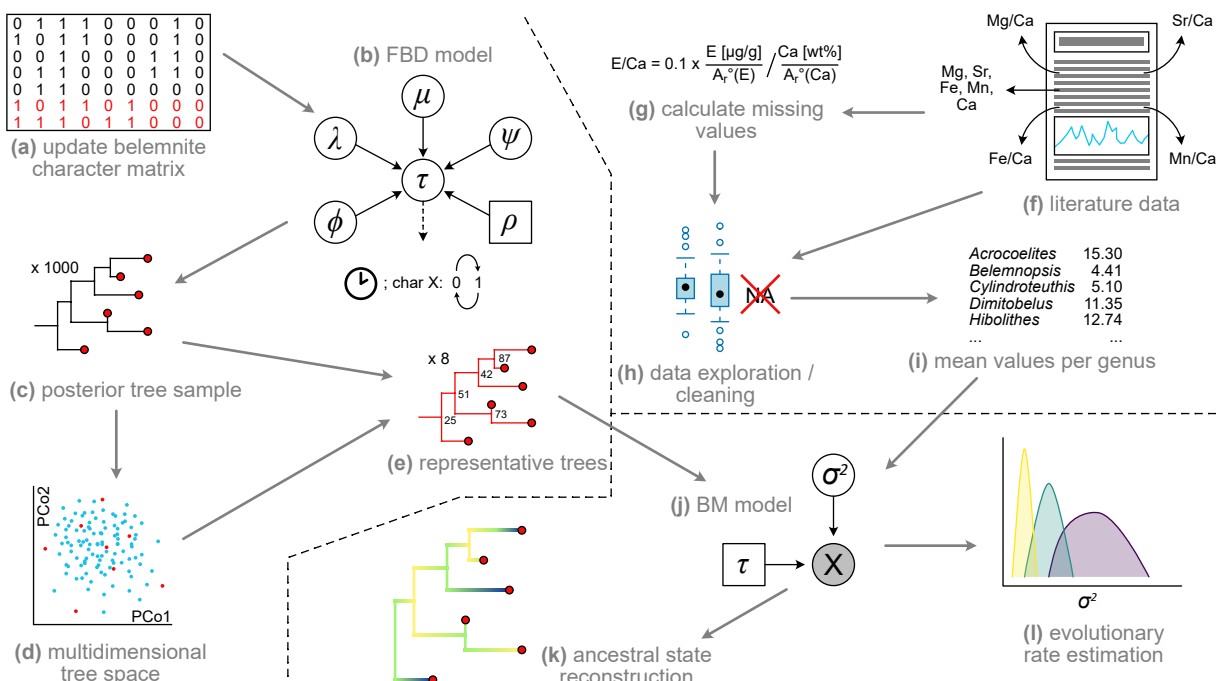

**Figure 1.** Workflow of the present study. **(a)** Collecting additional data for morphological character matrix to include taxa for which element ratio data are available. **(b)** Inference of tree topology ($\tau$) in RevBayes using the fossilised birth-death (FBD) model, consisting of extinction rate ($\mu$), speciation rate ($\lambda$), fossilisation rate ($\psi$), extant sampling rate ($\rho$) and origin time ($\phi$) in combination with the clock and character models (see Höhna et al., 2014 for details on graphical model representation). **(c)** Reduction of posterior trees to random sample of 1000 trees. **(d)** Multidimensional scaling (= principal coordinate analysis, PCo) of tree space to investigate topological differences between trees. **(e)** Choose representative trees from posterior tree sample, based on multidimensional scaling.**(f)** Literature survey, extracting element ratios and raw element concentrations. **(g)** Calculate element/Ca ratios where only raw values were given. **(h)** Remove non-informative values, compare element ratios between genera, localities, etc. **(i)** Calculate mean values for each genus. **(j)** Brownian Motion (BM) model consisting of the single parameter $\sigma^2$ and a fixed tree topology from previous steps. **(k)** Ancestral state reconstruction of element ratios in R. **(l)** Estimation of evolutionary rates in RevBayes.

of element/Ca ratios alone (e.g., Sørensen et al., 2015, ; see Discussion chapter herein). The latter data points also represented only a small fraction (less than 5%) of the complete dataset. Thus, despite some inherent potential for a diagenetic bias, our dataset is expected to represent an acceptable approximation of what might have been the element ratio distributions at the time of the secretion of the rostra. We also included a sensitivity test that randomly altered the original data to see whether slightly

biased data would affect the results (see below).

As expected, the reporting of elemental concentrations was inconsistent between the different sources used. Some authors reported all values of interest, while others reported raw values, relatively commonly omitting raw Ca content. For cases with all raw values known, we calculated the missing ratios as follows (Fig. 1g):

First, the molar amount of Ca per 1 g sample was calculated using the amount of Ca ($m_{Ca}$) in g, derived from the weight percentage of Ca ($w_{Ca}$) per sample, and the molar weight of Ca ($M_{Ca}$) according to the equation

$$n_{Ca}[mol] = \frac{m_{Ca}[g]}{M_{Ca}[\frac{g}{mol}]} \tag{1}$$

Second, the molar amount of Sr was calculated using the amount of Sr ($m_{Sr}$) in $\mu$g, derived from the concentration of Sr ($C_{Sr}$) in $\mu$g/g using the formula

$$n_{Sr}[mol] = \frac{m_{Sr}[\mu g] \times 10^{-6}}{M_{Sr}[\frac{g}{mol}]} \tag{2}$$

Third, the molar ratio of Ca to Sr (Sr/Ca) in mmol/mol was calculated according to

$$\frac{Sr}{Ca}[\frac{mmol}{mol}] = \frac{n_{Sr}[mol] \times 1000}{n_{Ca}[mol]} \tag{3}$$

The molar amount of Fe, Mn and Mg, as well as the molar ratios of Fe, Mn and Mg to Ca, were calculated accordingly.

Despite the fact that geochemical data have been collected from a large number of species, we restricted our analyses to the genus level (Fig. 1i). This is because we assume that the variation within the same genus is comparatively small. Moreover, this approach provides larger sample sizes for each taxon. We then inspected and cleaned this dataset according to several criteria (Fig. 1h): In most cases, identifications are given without any detailed justifications or images. Accordingly, it seems possible that the original authors included isolated cases of misidentifications. To mitigate this problem, we excluded specimens where the authors only tentatively assigned the specimens to a certain genus (but retained them when only the species level was uncertain). Large interspecific variation within genera could potentially cause biased average element ratios in these genera. However, there is currently no large belemnite phylogeny available that goes down to the species level and could be used as a basis for comparative phylogenetic studies. In any case, for many of the species reported in the literature, only a few data points are available, so results derived from a species-level analysis would be fraught with uncertainty. To ensure basic statistical robustness, we only included taxa with more than 10 specimens in the dataset. To assess the significance of basin and intra-basin effects, we compared data from the same section, where several different genera were reported. Lastly, we also compared data from genera that were reported from multiple localities.

**Table 2.** Literature sources for belemnite geochemical data. Abbreviations of genus names: *Acc.* = *Acrocoelites*, *Act.* = *Acroteuthis*, *Adi.* = *Adiakritobelus*, *Aul.* = *Aulacoteuthis*, *Bai.* = *Bairstowius*, *Ber.* = *Berriasibelus*, *Blp.* = *Belemnopsis*, *Blt.* = *Belemnites*, *Blx.* = *Belemnellocamax*, *Brv.* = *Brevibelus*, *Con.* = *Conohibolites*, *Cst.* = *Castellanibelus*, *Cyl.* = *Cylindroteuthis*, *Dim.* = *Dimitobelus*, *Duv.* = *Duvalia*, *Hib.* = *Hibolithes*, *Hlc.* = *Holcobelus*, *Hst.* = *Hastites*, *Lag.* = *Lagonibelus*, *Mes.* = *Mesohibolites*, *Mir.* = *Mirabelobelus*, *Nan.* = *Nannobelus*, *Neo.* = *Neohibolithes*, *Oxy.* = *Oxyteuthis*, *Pas.* = *Passaloteuthis*, *Per.* = *Peratobelus*, *Pox.* = *Praeoxyteuthis*, *Ppa.* = *Parapassaloteuthis*, *Psb.* = *Pseudobelus*, *Psd.* = *Pseudoduvalia*, *Sim.* = *Simpsonibelus*, *Vau.* = *Vaunagites*, *Yng.* = *Youngibelus*

| Reference | Geography | Stratigraphy | Genera | Elements |
|---|---|---|---|---|
| Alberti et al. (2021a)[a] | Indian Himalaya | Callovian-Tithonian | *Blp.*? | Ca, Mg, Sr, Fe, Mn |
| Alberti et al. (2021b) | Neuquén Basin, Argentina | Bajocian-Tithonian | *Blp.*, *Brv.*, *Hib.* | Ca, Mg, Sr, Fe, Mn |
| Arabas (2016) | Carpathians, Poland / Slovakia | Aalenian-Tithonian | *Blp.*, *Duv.*, *Hib.* | Ca, Mg, Sr, Fe, Mn |
| Arabas et al. (2017) | Carpathians, Ukraine / Slovakia | Pliensbachian, Aalenian | *Acc.*, *Hlc.*, *Ppa.*, *Pas.* | Ca, Mg, Sr, Fe, Mn |
| Armendáriz et al. (2012)[a] | Asturian Basin, Spain | Pliensbachian | *Pas.*? | Ca, Mg, Sr, Fe, Mn |
| Benito and Reolid (2012)[a] | Betic Cordillera, Spain | Oxfordian-Kimmeridgian | *Hib.*? | Ca, Mg, Sr, Fe, Mn |
| Dutton et al. (2007) | Seymour Island, Antarctica | Maastrichtian | *Dim.* | Ca, Mg, Sr, Fe, Mn |
| Li et al. (2012) | Yorkshire coast, England, UK | Toarcian | *Acc.*, *Sim.*, *Yng.* | Ca, Mg, Sr, Fe, Mn |
| Li et al. (2013) | Dorset coast, England, UK / Vocontian Basin, France | Pliensbachian, Callovian, Valanginian | *Bai.*, *Ber.*, *Cst.*, *Cyl.*, *Duv.*, *Hst.*, *Hib.*, *Nan.*, *Pas.* | Ca, Mg, Sr, Fe, Mn |
| Malkoč et al. (2010)[a] | Hannover, Germany | Barremian-Aptian | *Aul.*?, *Neo.*?, *Oxy.*?, *Pox.*? | Ca, Mg, Sr, Fe, Mn |
| McArthur et al. (2004) | Yorkshire coast, England, UK | Valanginian-Barremian | *Act.*, Aul., *Hib.*, *Oxy.*, *Pox.* | Ca, Mg, Sr, Fe, Mn |
| McArthur et al. (2007) | Betic Cordillera, Spain / Vocontian Basin, France | Berriasian-Hauterivian | *Adi.*, *Ber.*, *Cst.*, *Duv.*, *Hib.*, *Mir.*, *Psb.* | Ca, Mg, Sr, Fe, Mn |
| Pirrie et al. (2004)[b] | Lago San Martin, Argentina | Albian | *Dim.* | Mg, Sr, Fe, Mn |
| Price and Mutterlose (2004)[b] | Yatria River, Western Siberia, Russia | Berriasian-Hauterivian | *Act.*, *Cyl.*, *Lag.* | Mg, Sr, Fe, Mn |
| Price et al. (2009)[b] | Wiltshire, England, UK | Oxfordian-Kimmeridgian | *Cyl.* | Mg, Sr, Fe, Mn |

| Reference | Geography | Stratigraphy | Genera | Elements |
|---|---|---|---|---|
| Price et al. (2011) | Gerecse Mountains, Hungary | Valanginian-Barremian | *Adi.*, "*Blt.*", *Con.*, *Duv.*, *Hib.*, *Mes.*, *Psb.*, *Psd.*, *Vau.* | Ca, Mg, Sr, Fe, Mn |
| Price et al. (2012) | Carnarvon & Eromanga Basins, Australia | Barremian-Cenomanian | *Dim.*, *Per.* | Ca, Mg, Sr, Fe, Mn |
| Sørensen et al. (2015) | Kristianstad Basin, Sweden | Campanian | *Blx.* | Ca, Mg, Sr, Mn |
| Stevens et al. (2014) | Nusplingen, Germany | Kimmeridgian | *Hib.* | Ca, Mg, Sr, Fe, Mn |
| Stevens et al. (2017) | Münsterland, Germany | Albian | *Neo.* | Ca, Mg, Sr, Fe, Mn |
| Stevens et al. (2022) | Polski Trambesh, Bulgaria | Aptian | *Duv.*, *Mes.* | Ca, Mg, Sr, Fe, Mn |
| Ullmann et al. (2013) | NW North Island, New Zealand | Kimmeridgian-Tithonian | *Blp.*, *Hib.* | Ca, Sr, Mn |
| Ullmann et al. (2014) | Yorkshire coast, England, UK | Toarcian | *Acc.*, *Ppa.*, *Pas.*, *Yng.* | Ca, Mg, Sr, Mn |
| Ullmann et al. (2015) | Yorkshire coast, England, UK | Toarcian | *Pas.* | Ca, Mg, Sr, Mn |
| Ullmann and Pogge von Strandmann (2017) | Yorkshire coast, England, UK | Toarcian | *Pas.* | Ca, Mg, Sr, Mn |
| van de Schootbrugge et al. (2000)[b] | Vocontian Basin, France | Valanginian-Hauterivian | *Ber.*, *Cst.*, *Duv.*, *Hib.*, *Psb.* | Mg, Sr, Fe, Mn |
| Vickers et al. (2021) | Christian Malford, England, UK | Callovian | *Cyl.* | Ca, Mg, Sr, Fe, Mn |
| Wierzbowski and Joachimski (2007)[b] | Jura Chain, Poland | Bajocian-Bathonian | *Blp.*, *Hib.* | Mg, Sr, Fe, Mn |
| Wierzbowski and Rogov (2011) | Volga Basin, SW Russia | Callovian-Oxfordian | *Cyl.*, *Hib.* | Ca, Mg, Sr, Fe, Mn |

[a] Generic affinity uncertain for all taxa in publication, no data included in our analyses.

[b] Only raw element concentrations given, but not for Ca, no element ratios calculated.

## 2.2 Morphology-based phylogeny

We used the morphological character matrix from Stevens et al. (2023) as a basis for our analyses (Fig. 1a). Several genera for which element ratios are available are missing from that phylogeny. Consequently, we updated the character matrix and ran a new analysis to include the missing taxa. The additional taxa and the corresponding references are listed in Ta-

ble 3. We removed aulacoceratids (*Atractites*, *Aulacoceras* and *Palaeobelemnopsis*) and sinobelemnitids (*Sinobelemnites* and *Tohokubelus*) from the analysis because geochemical data have not been reported for any of these taxa. Aulacoceratids also have a primary aragonitic rather than calcitic rostrum (Dauphin and Cuif, 1980). Both groups are considered to be ancestral to other belemnitids: aulacoceratids are likely stem coleoids (Mariotti et al., 2021; Hoffmann et al., 2022; Stevens et al., 2023), while sinobelemnitids are probably paraphyletic, basal to other belemnitids (Iba et al., 2012; Niko and Ehiro, 2022; Stevens

et al., 2023). In the original character matrix, *Neohibolites minimus* was scored as 'absent' for the primary character 'ventral alveolar furrow(s)' but included secondarily dependent characters that were not scored as 'inapplicable'. This (unintentional) character scoring represents a logically impossible character state combination (see Sereno, 2007) and was therefore corrected. Accordingly, we changed the primary character to 'present'. Otherwise, we left the character coding and scoring unchanged, apart from the addition of new species and the removal of phylogenetically uninformative characters (i.e., characters that were

only relevant for sinobelemnitids and aulacoceratids).

The morphological character matrix was analysed with Bayesian phylogenetic inference using the software REVBAYES 1.2.1 (Höhna et al., 2016). The following parameter settings were therefore relevant in a first step of the study to recover trees without involving geochemical data (Fig. 1b) and roughly follow what was used in Stevens et al. (2023). We supply brief explanations for each model component but refer the reader to the literature for further methodological background (e.g., Wright, 2019;

Wright et al., 2022). We applied the fossilised birth-death (FBD) model (Stadler, 2010; Gavryushkina et al., 2014; Heath et al., 2014) with exponential priors on extinction and speciation rate with lambda = 10, respectively, the same for the fossil recovery rate. Extinction and speciation rates here model the underlying diversification process, while the fossil recovery rate represents sampling in the FBD model. Another exponential prior with lambda = 0.1 and an offset of 201.4 was put on origin time (i.e., the time where the diversification process started), corresponding to the appearance of the oldest belemnite in our sample and the

oldest known non-sinobelemnitid belemnite in general, *Schwegleria feifeli* (Schlegelmilch, 1998, Hettangian). Priors represent the distributions from which model parameters are drawn. The exclusively binary morphological characters were modelled using the Mkv model (Lewis, 2001), which represents a simple model of morphological evolution where the transition rates between character states (e.g., loss or gain of a structure) are assumed to be equal. As not all characters are expected to evolve at the same rate, we used a discretised Gamma distribution with four rate categories to model rate heterogeneity across characters.

A lognormal morphological clock was then applied to account for variable rates across branches, i.e., the morphological rates also vary between different lineages. Finally, to account for uncertainties in the stratigraphic ages of the taxa, we applied operators for variable tip dates (Barido-Sottani et al., 2019, 2020). This provides minimum and maximum age constraints for each taxon in the analysis. Importantly, because sampling is modelled as a single event for each taxon, a different age (complying with these constraints) of the taxon was drawn for each iteration of the algorithm. With all of these settings, we ran

four independent replicates with 100,000 generations, each using a random move schedule, sampling every 10 generations and discarding 25% of the samples as burn-in. Convergence was assessed using TRACER 1.7.2 (Rambaut et al., 2018) by visual inspection and checking that all parameters reached effective sampling sizes above 200.

The result of a Bayesian phylogenetic analysis is a large sample of trees, which is non-trivial to summarise (e.g., Heled and Bouckaert, 2013; O'Reilly and Donoghue, 2018; Berling et al., 2024). Ancestral state estimation aims to reconstruct the

**Table 3.** List of additional species for the updated phylogeny, with references used to collect morphological data.

| Species | Family | Age | References |
|---|---|---|---|
| *Acroteuthis arctica* Blüthgen, 1936 | Cylindroteuthidae | Berriasian-Valangian | Stolley (1911); Sachs and Nal'nyaeva (1966) |
| *Bairstowius junceus* (Phillips, 1867) | Hastitidae | Late Pliensbachian | Doyle (1994); Schlegelmilch (1998) |
| *Belemnellocamax mammillatus* (Nilsson, 1826) | Belemnitellidae | Late Campanian | Christensen (1975, 1997) |
| *Berriasibelus exstinctorius* (Raspail, 1829) | Duvaliidae | Early Valangian | Combémorel (1973); Janssen (2003) |
| *Brevibelus breviformis* (Voltz, 1830) | Acrocoelitidae | Late Toarcian | Schlegelmilch (1998) |
| *Castellanibelus orbignyanus* (Duval-Jouve, 1841) | Duvaliidae | Early Valangian | Combémorel (1973) |
| *Conohibolites escargollensis* (Delattre, 1952) | Belemnopseidae | Late Barremian | Janssen and Főzy (2005) |
| *Lagonibelus gustomesovi* Sachs & Nal'nyaeva, 1964 | Cylindroteuthidae | Early Berriasian | Sachs and Nal'nyaeva (1964) |
| *Nannobelus acutus* (Miller, 1826) | Passaloteuthidae | Late Sinemurian | Schlegelmilch (1998) |
| *Peratobelus oxys* Tenison-Woods, 1883 | Dimitobelidae | Aptian | Williamson (2006) |
| *Praeoxyteuthis pugio* (Stolley, 1925) | Oxyteuthidae | Early Barremian | Mutterlose (1983) |
| *Pseudobelus brevis* Paquier, 1900 | Duvaliidae | Hauterivian | Combémorel (1973) |
| *Simpsonibelus dorsalis* (Phillips, 1867) | Acrocoelitidae | Toarcian | Doyle (1991); Schlegelmilch (1998) |
| *Youngibelus simpsoni* (Mayer-Eymar, 1883) | Acrocoelitidae | Early Toarcian | Doyle (1991); Schlegelmilch (1998) |

evolutionary changes of a trait along a given tree. This is typically performed on a single fixed tree, which may not be a good representative of the tree sample. On the other hand, performing ancestral state reconstruction on a larger sample of trees is challenging to interpret, as a summary tree with mapped character changes can include reconstructions from very different topologies. For these reasons, we opted for a middle ground between these approaches, using different trees from the posterior sample that are representative of (i) the entire sample, (ii) identified clusters of the multidimensional tree space, and (iii) the extremes of the sampled tree space. To select these trees, we used the package TREESPACE 1.1.4.2 (Jombart et al., 2017) in R 4.3.0 (R Core Team, 2018). For computational reasons, we selected a random sample of 1000 trees from the posterior trees (Fig. 1c). Pairwise distances were calculated for these trees using the quartet distance metric in the package QUARTET 1.2.6 (Smith, 2019), which overcomes some of the limitations of several other tree metrics (Smith, 2020, 2022). Examples include the Kendall-Colijn metric (Kendall and Colijn, 2016) and the Robinson-Foulds distance (Robinson and Foulds, 1979, 1981), which are implemented as a standard in TREESPACE (Jombart et al., 2017). Analogous to functions in TREESPACE, the resulting distance matrix was first transformed into a Euclidean distance matrix using the package ADE4 1.7.22 (Dray and Dufour, 2007).

We then performed metric multidimensional scaling (= principal coordinate analysis) on this distance matrix (Fig. 1d), as implemented in TREESPACE and identified clusters of trees (denoted as A, B, C) using Ward's method (Ward, 1963). To compare the distribution of clade supports among these clusters, we used the package APE 5.7.1 (Paradis and Schliep, 2019). We calculated the proportion of trees that contained an identical clade (i.e., the posterior probability of the clade, PP) for every clade in every tree and the average value for each tree. We additionally calculated partial posterior probabilities of clades, where we only considered trees that belong to the same cluster as identified above. These partial posterior probabilities are then compared to the total posterior probability of the clade to gain insights into what degree the tree clusters supported certain topologies. Posterior clade probabilities of the total posterior tree sample are abbreviated as PPtot, and partial posterior probabilities of the corresponding clusters are $PP_A$, $PP_B$, $PP_C$, respectively.

After exploring the tree landscape as described above, we selected eight trees as representatives for our subsequent analyses of ancestral state reconstruction and evolutionary rates (Fig. 1e):

1. The median tree of the entire tree sample (see Jombart et al., 2017).

2. The maximum clade credibility (MCC) tree of cluster A.

3. The MCC tree of cluster B. This tree coincides with the MCC tree from the entire tree sample.

4. The MCC tree of cluster C.

5. The tree with the smallest value for principal coordinate axis 1.

6. The tree with the highest value for principal coordinate axis 1.

7. The tree with the smallest value for principal coordinate axis 2.

8. The tree with the highest value for principal coordinate axis 2.

Branch lengths were not rescaled but kept at the values of the original trees, i.e., the timing of branching events (nodes) in these trees are not averaged over all trees. Instead, each tree represents a single sample from the posterior.

### 2.3 Ancestral states and evolutionary rates

Fe/Ca and Mn/Ca ratios did not yield reliable data, as the values were commonly below the detection limits (the detection limit varied depending on the study cited); therefore, we only used Mg/Ca and Sr/Ca ratios for the analysis of ancestral states and evolutionary rates (Fig. 1j-l). As input for each taxon, we used the median of the element ratio within the genus. Taxa that did not have any data for the corresponding element ratios were removed prior to the analyses (e.g., *Megateuthis* for both element ratios or *Holcobelus* for Sr/Ca). Consequently, the analyses for Mg/Ca and Sr/Ca ratios were based on a tree with 25 and 24 taxa, respectively.

We reconstructed ancestral states (Fig. 1k) using the 'fastAnc' function in the R-package PHYTOOLS 2.1.1 (Revell, 2024). For visualisation, we also plotted a phenogram using the same package. The latter plots the continuous trait values of all taxa

on the y-axis, and time on the x-axis. The taxa are then connected by their phylogenetic relationships, as indicated in the corresponding tree. For the sake of brevity, we only used the overall MCC tree as input for this analysis.

The estimation of evolutionary rates was performed in REVBAYES 1.2.1 (Höhna et al., 2016), with one analysis fixing each of the eight trees mentioned in the previous section (Fig. 1l). To all combinations of the trees and the two element ratios, we applied a simple evolutionary model of Brownian motion (Felsenstein, 1985), with the rate $\sigma^2$ as the only parameter (Fig. 1j). Thus, for simplicity, the evolutionary rate was assumed to be constant throughout the tree. We placed a log-uniform prior between 10 and 100 on $\sigma^2$ to allow for a wide range of values. For each analysis, we ran two independent replicates using a random move schedule for 50,000 generations after a burn-in of 5,000 generations, sampling every 10 generations. As for the tree inference, convergence was checked using TRACER (Rambaut et al., 2018).

Although a step forward from the inference based on a single tree, the above-described eight trees still only reflect point estimates of the tree space. For this reason, we repeated the same procedure for the random subsample of posterior trees from the tree space analysis, resulting in 2000 additional analyses (note that the exemplar trees are contained within the posterior tree sample; 16 analyses were, therefore, technically duplicated). We compared the distribution of $\sigma^2$ of Mg/Ca and Sr/Ca in the exemplar trees to its distribution across all trees. We also tested whether different tree clusters implied different evolutionary rates. The advantage of having analyses based on a small number of trees alongside the full tree sample is that it allows for a more direct comparison with tree topology while at the same time accounting for uncertainty across tree space.

To test the sensitivity of our models, we applied another test where the primary data (i.e., the median values of Mg/Ca and Sr/Ca for each taxon) were randomly altered. For the list of median values of Mg/Ca and Sr/Ca per taxon, we applied the following R commands to retrieve randomly altered values:

```
abs(Mg_ratio + (sample(c(-1, 1), 1) * rexp(length(Mg_ratio), 1)))
abs(Sr_ratio + (sample(c(-1, 1), 1) * rexp(length(Sr_ratio), 10)))
```

This means that a random number was drawn from an exponential distribution and either added or subtracted (with equal probability) from the original value. The rate parameter of the exponential distribution was 1 for the Mg/Ca ratio and 10 for the Sr/Ca ratio, i.e., with expected mean values of 1 and 0.1, respectively. Thus, the median values for each taxon were altered on average by the latter values, which roughly corresponds to 10% of the mean of these element ratios (see Table 4). This is intended to model a situation where some of the data may be affected by diagenetic alteration, with most of the data being close to the original data, but a some data points changed by a larger amount. We repeated this process 100 times for each element and ran the analyses with the same settings as above, fixing the MCC tree of the entire sample. These results were then compared with the results from the analysis of cluster B's MCC tree, which used the unaltered data on the same tree.

**Table 4.** Statistical summary of element ratios of entire dataset.

| Ratio | n | min | 25%Q | median | mean | 75%Q | max |
|---|---|---|---|---|---|---|---|
| Mg/Ca | 2152 | 1.527 | 8.403 | 10.398 | 10.750 | 12.943 | 28.100 |
| Sr/Ca | 2241 | 0.392 | 1.387 | 1.520 | 1.597 | 1.762 | 3.300 |
| Mn/Ca | 1585 | -0.010[a] | 0.007 | 0.013 | 0.091 | 0.041 | 4.580 |
| Fe/Ca | 1062 | 0.000[a] | 0.011 | 0.040 | 0.679 | 0.141 | 26.668 |

[a] Ratios are taken directly from original publications, likely representing values below detection threshold, but not referenced as such. Included in calculations here, though neither ancestral state analyses nor evolutionary rate estimations were performed for Mn/Ca and Fe/Ca.

## 3 Results

### 3.1 Element/Ca ratio data

In total, we compiled element ratio data from 29 published studies (Table 2). Four studies were discarded because they contained only taxa with uncertain generic assignments or genera with less than 10 specimens in the entire dataset. Five studies could not be used because the authors neither supplied element ratios nor raw Ca concentrations. Without these data, we were unable to calculate element ratios. The total dataset included 2749 individual data points. After excluding the studies mentioned above and data as specified in the methods, our dataset included 2241 data points.

A statistical summary of the element ratios is given in Table 4. For all four element ratios, belemnite genera showed distinct, although partly overlapping distributions (Fig. 2). The total ranges of element ratios within each genus were relatively wide, especially where rostra had been sampled ontogenetically (e.g., Ullmann et al., 2013, 2015; Sørensen et al., 2015; Ullmann and Pogge von Strandmann, 2017; Stevens et al., 2017, 2022). The second quartile and third quartile (i.e., the central 50% of the data) were relatively narrow for Mg/Ca and Sr/Ca in most genera.

In the case of Fe/Ca and Mn/Ca, extremely positive outliers were much more common. The mean skewness of the element ratio data per genus amounts to 0.42 for Mg/Ca, 0.31 for Sr/Ca, 2.39 for Fe/Ca, and 3.12 for Mn/Ca. A skewness of 0.0 represents a balanced distribution, while negative values indicate left skew and positive values right skew. Higher values indicate an increasingly longer tail of the distribution. The skewness becomes even stronger when considering that many values were excluded because they were below the detection limits. Thus, a large portion of the Fe/Ca and Mn/Ca ratios were close to zero. These element ratios were not used in the remainder of the analyses because the skew may reduce the representativeness of mean or median values for the genera. In addition, the Fe/Ca and Mn/Ca ratios were more homogeneous between genera; this induces difficulty when aiming to recognise a phylogenetic pattern.

Distributions of element ratios had strong overlap within species of the same genus (Fig. 3). Specimens in open nomenclature (i.e., only determined to genus level) showed a wider range of Mg/Ca ratios than identified species in *Hibolithes* (Fig. 3a) and *Duvalia* (Fig. 3b). In the latter case, the average Mg/Ca ratios were distinctly lower than in identified species of the genus. The pattern in *Passaloteuthis* shows a clear bimodal distribution, with *Passaloteuthis bisculata*, *P. elongata*, *P. cuspidatus*

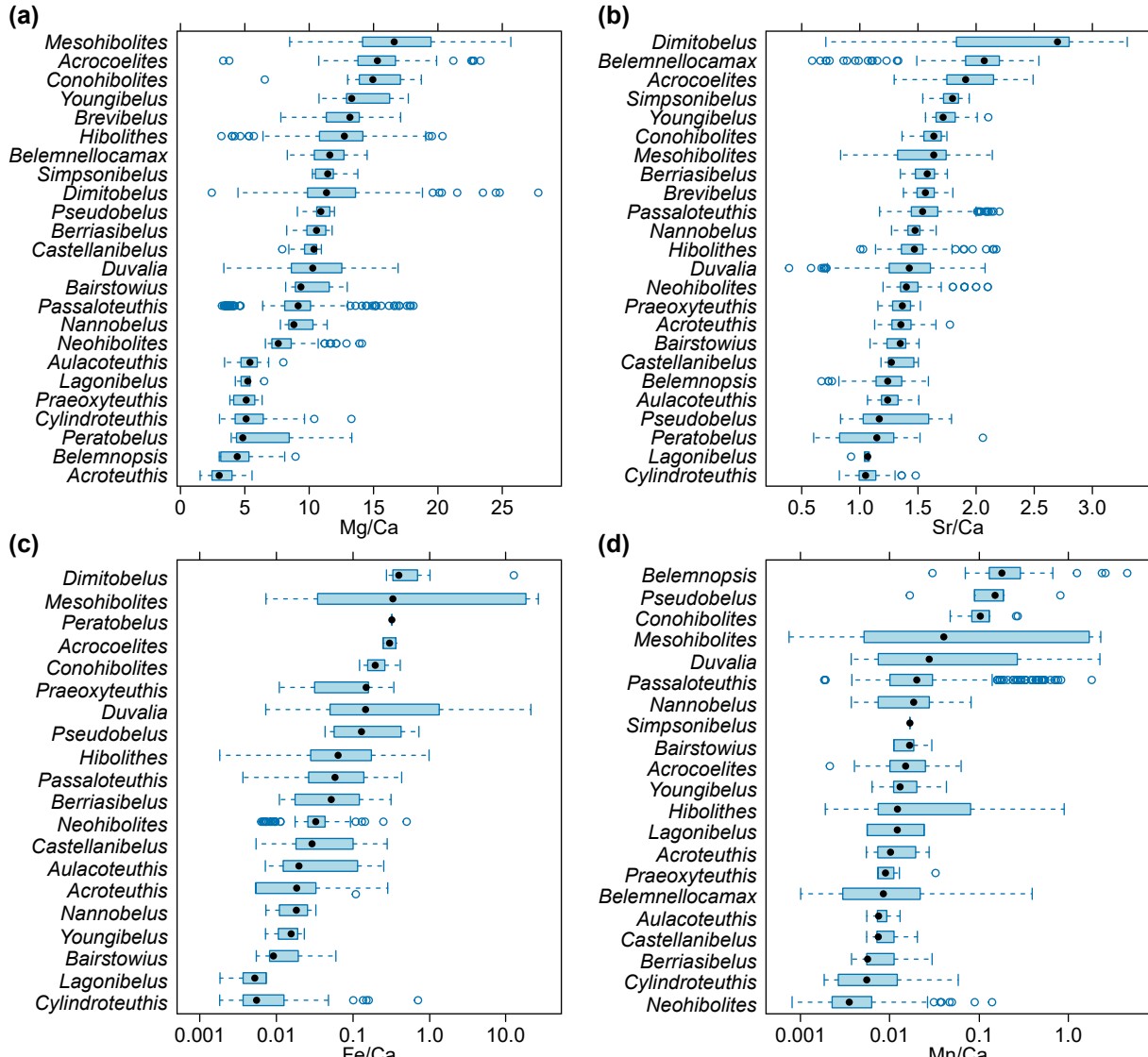

**Figure 2.** Belemnite rostrum element ratio distributions per genus. **(a)** Mg/Ca ratios. **(b)** Sr/Ca ratios. **(c)** Fe/Ca ratios, log scale. **(d)** Mn/Ca ratios, log scale. Data taken from entire literature dataset, see text for references.

and specimens in open nomenclature containing higher Mg/Ca ratios than other species (Fig. 3c). Note that *Passaloteuthis* specimens with lower Mg/Ca ratios came from the same study that includes data on the early to late Pliensbachian at the locality Priborzhavske in Ukraine (Arabas et al., 2017). All other data for *Passaloteuthis* come from two localities in the UK, spanning the Pliensbachian and early Toarcian (Li et al., 2013; Ullmann et al., 2014, 2015; Ullmann and Pogge von Strandmann, 2017). It is thus possible that some of the identifications were incorrect or that there were other factors involved in the anomalous values of the former study.

The distributions of Mg/Ca ratios were taxon-specific within the same locality and congruent between localities (Fig. 4). This pattern was independent of the stratigraphic position. For example, at Dubki (Russian Platform), *Hibolithes* rostra occur throughout the section but do not overlap in their distribution of Mg/Ca ratios with the co-occurring *Lagonibelus* and *Cylindro-*

270 *teuthis* (data from Wierzbowski and Rogov, 2011). Conversely, specimens of *Hibolithes*, *Duvalia* and *Belemnopsis* at Stankowa Skała (Poland) from several stratigraphic positions have almost identical Mg/Ca ratios (data from Arabas, 2016).

## 3.2  Morphology-based phylogeny

Overall, our morphological phylogenetic analyses recovered similar topologies as in Stevens et al. (2023), although the topological uncertainties were generally high. Multidimensional scaling revealed three clusters of trees, here termed clusters A, B and C (Fig. 5). The first two clusters make up the largest portion of tree space, with 41% of the trees falling into cluster A and

275 49% into cluster B, the remaining 10% make up cluster C. The boundaries between the clusters are gradual in all investigated PCo axes. It is notable that cluster C is less dense and occupies an intermediate position between the other clusters in PCo axis 1 (Fig. 5b, c). This implies that clusters A and B have higher posterior densities and represent opposite optimal islands. Transitional topologies are represented by cluster C and are less frequent. In PCo axis 2 (Fig. 5b, d), all three clusters overlap; therefore, this axis does not account for differences between the clusters. PCo axis 3 shows a separation of cluster C from the

280 other two clusters (Fig. 5c, d). The average posterior probabilities across all clades calculated for each tree did not fall outside the interval between 0.1 and 0.4. Trees in clusters A and B have higher average posterior probabilities due to a higher density of similar trees (Fig. 5a).

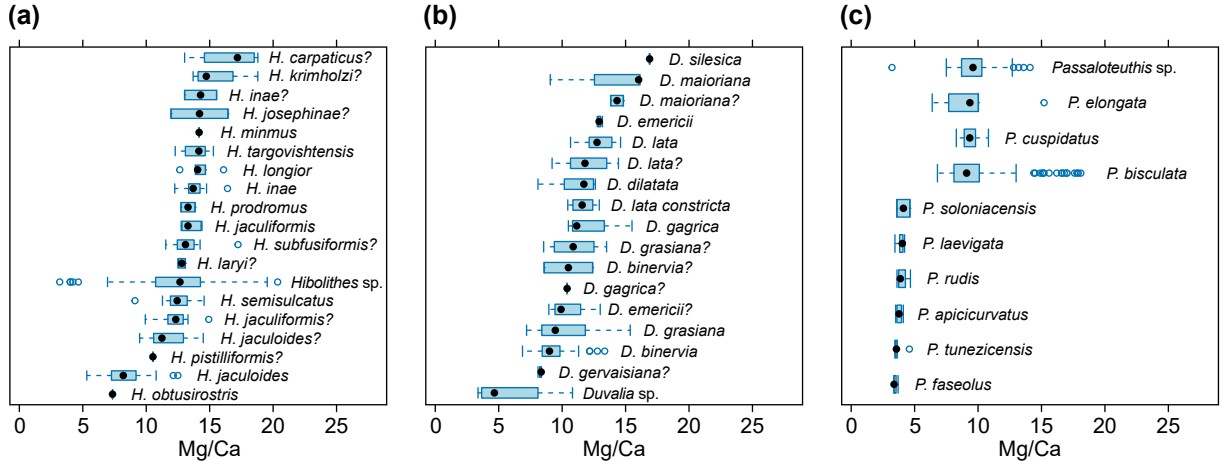

**Figure 3.** Belemnite rostrum Mg/Ca ratios of species within the same genus. **(a)** Species of *Hibolithes*, Callovian to Barremian (van de Schootbrugge et al., 2000; McArthur et al., 2004, 2007; Wierzbowski and Joachimski, 2007; Price et al., 2011; Li et al., 2013; Ullmann et al., 2013; Stevens et al., 2014; Arabas, 2016; Alberti et al., 2021b). **(b)** Species of *Duvalia*, Oxfordian to Aptian (van de Schootbrugge et al., 2000; McArthur et al., 2007; Price et al., 2011; Li et al., 2013; Arabas, 2016; Stevens et al., 2022). **(c)** Species of *Passaloteuthis*, Pliensbachian to Toarcian (Li et al., 2013; Ullmann et al., 2014, 2015; Arabas et al., 2017; Ullmann and Pogge von Strandmann, 2017).

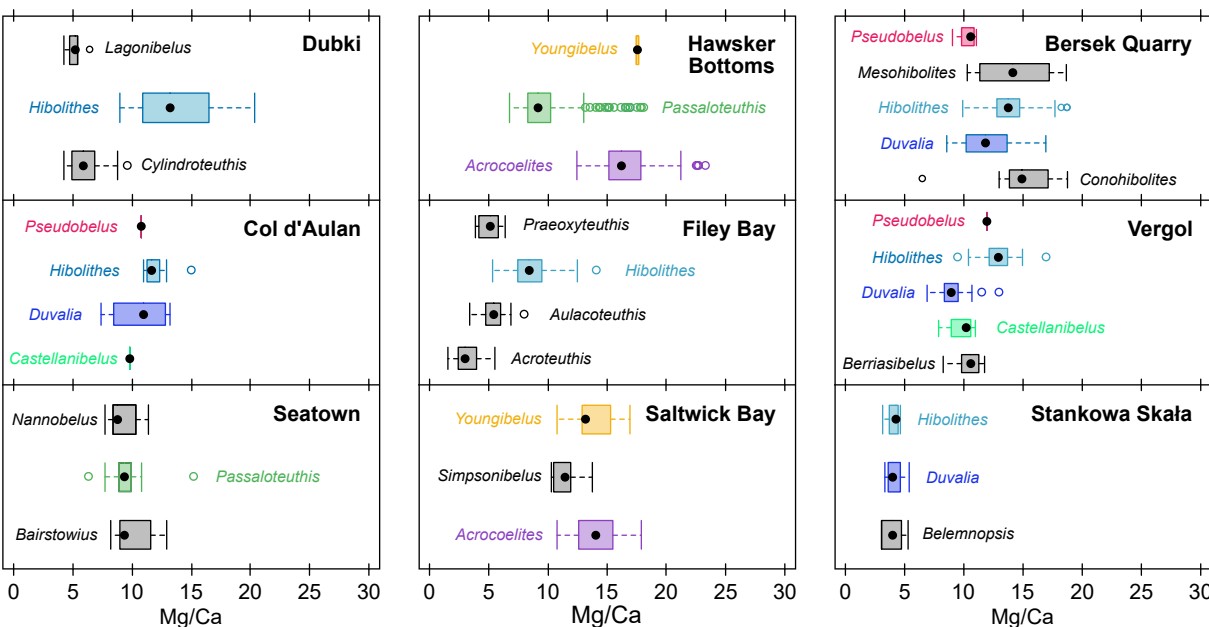

**Figure 4.** Mg/Ca ratios per genus within the same locality. Colours represent unique genera that occur in more than a single locality. Genera in black only occur once in this comparison. Note that we do not account for time averaging here, and individual genera may show non-overlapping stratigraphic ranges. Localities: Dubki, Russian Platform, late Callovian to early Oxfordian (Wierzbowski and Rogov, 2011); Col d'Aulan, Southeastern France, early to late Valanginian (McArthur et al., 2007); Seatown, Southern England, Pliensbachian (Li et al., 2013); Hawsker Bottoms, Northern England, Early Toarcian (Ullmann et al., 2014, 2015; Ullmann and Pogge von Strandmann, 2017); Filey Bay, Northern England, Valanginian to Barremian (McArthur et al., 2004); Saltwick Bay, Northern England, Early Toarcian (Li et al., 2012; Ullmann et al., 2014); Bersek Quarry, Northern Hungary, late Valanginian to Barremian (Price et al., 2011); Vergol, Southeastern France, early to late Valanginian (McArthur et al., 2007; Li et al., 2013). Stankowa Skała, Southern Poland, late Oxfordian to early Kimmeridgian (Arabas, 2016).

The selected exemplar trees provide an overview of the topological variation between tree clusters (Fig. 6). For ease of comparison, we designated four monophyletic clades in the general MCC tree (which is equivalent to the MCC tree from cluster B; Fig. 6b):

1. Pseudoalveolata (see definition in Stevens et al., 2023), $PP_{tot} = 0.64$.

2. Belemnitina, including Megateuthidae and Passaloteuthidae, $PP_{tot} = 0.06$.

3. A clade here referred to as DuvPO, comprising Duvaliidae, *Praeoxyteuthis* and *Oxyteuthis*, $PP_{tot} = 0.44$.

4. A clade here referred to as CylABY, encompassing Cylindroteuthidae, *Aulacoteuthis*, *Bairstowius* and *Youngibelus*, $PP_{tot} = 0.16$.

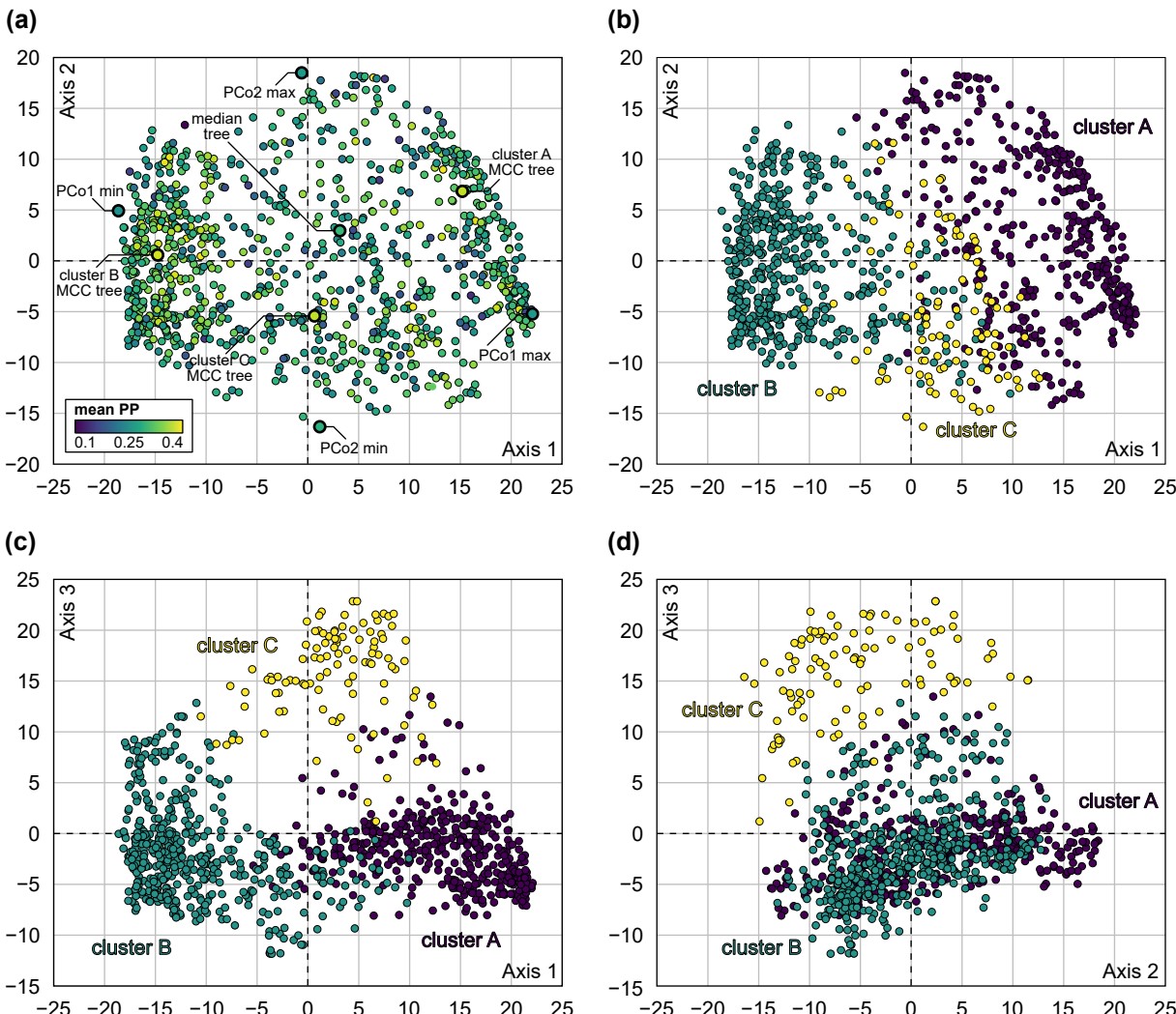

**Figure 5.** Principal coordinate analysis (PCo, multidimensional scaling) of tree space of subsample of 1000 trees, based on quartet distance. **(a)** PCo axes 1 and 2, coloured by mean posterior probability of corresponding tree, with position of exemplar trees highlighted. **(b)** PCo axes 1 and 2, coloured by tree cluster. **(c)** PCo axes 1 and 3, coloured by tree cluster. **(d)** PCo axes 2 and 3, coloured by tree cluster.

Note that except for the Pseudoalveolata, we do not consider any of these clades as definite taxonomic units due to the high uncertainties. Rather, we use them to facilitate comparison with alternative topological configurations across tree space. The posterior support for the Pseudoalveolata was variable between tree clusters ($PP_A = 0.83$; $PP_B = 0.53$; PPC = 0.37; PPtot = 0.64). This is partly caused by the uncertainty in the position of the DuvPO clade, which was occasionally recovered as forming part of the Pseudoalveolata ($PP_{tot} = 0.16$). Alternative positions of DuvPO included a sister group relationship with some of the Cylindroteuthidae ($PP_{tot} = 0.18$), which was more prevalent in tree cluster A, resulting in $PP_A = 0.43$ for the same clade (Fig. 6a).

Likewise, in cluster A, CylABY was paraphyletic, but a monophyletic clade uniting the Cylindroteuthidae and DuvPO received higher support than in the total tree sample ($PP_A = 0.48$; $PP_{tot} = 0.19$). DuvPO was paraphyletic in tree cluster C (Fig. 6c), with *Praeoxyteuthis* and *Oxyteuthis* forming a monophyletic clade together with the Pseudoalveolata ($PP_C = 0.70$; $PP_{tot} = 0.10$). Therefore, the two genera occupied a more nested position within the Pseudoalveolata rather than being its sister group (i.e., monophyletic Pseudoalveolata: $PP_C = 0.37$) in about half of the trees in cluster C. Concurrently, the Duvaliidae received considerably higher support in cluster C than in the total tree sample ($PP_C = 0.80$; $PP_{tot} = 0.44$). The support values in tree cluster C thus differ strongly from the total posterior tree sample. However, cluster C accounts only for a small proportion of the trees; thus, deviations in clade probabilities are to be expected. In comparison with tree clusters A and B, the MCC tree of cluster C contains several clades which are not supported by the total tree sample. These clades had posterior probabilities below 0.01, which means that out of 1000 trees, less than 10 trees shared an identical clade. The Cylindroteuthidae were more frequently monophyletic in tree cluster B ($PP_B = 0.78$; $PP_{tot} = 0.57$), which also applied to the CylABY, although with higher uncertainty ($PP_B = 0.23$; $PP_{tot} = 0.16$).

As the median tree was part of cluster C, its topological configurations resembled the MCC tree of cluster C (Fig. 6d). The clades in the median tree had even lower posterior probabilities, with 10 out of 35 nodes having posterior probabilities of 0.01 or below. As a comparison, the MCC tree from cluster C contained three nodes with $PP_{tot} \leq 0.01$, while the MCC trees from clusters A and B contained none. This suggests that the median tree is a very poor point estimate of the general tree topology despite its central position within the tree space.

The trees representing the limits of tree space (Fig. 6e-h) had a generally similar arrangement of the clades as the MCC trees of their respective clusters. However, the number of barely supported clades was higher (between 5 and 9 nodes with $PP_{tot} \leq 0.01$). Accordingly, the maximum trees of the PCo1 and PCo2 axes, as part of cluster A, contained a monophyletic clade containing a paraphyletic DuvPO and part of the CylABY. This clade was more closely related to the Belemnitina in the PCo1 maximum tree with negligible support ($PP_{tot} = 0.01$). In the PCo2 maximum tree, this mixed DuvPO-CylABY clade formed a monophyletic group together with *Belemnopsis* as the sister group to the Pseudoalveolata ($PP_{tot} = 0.01$). Note that the latter tree recovered *Cylindroteuthis* itself elsewhere. As part of cluster B, the minimum tree of the PCo1 axis recovered a mono-phylum uniting DuvPO and Pseudoalveolata. In contrast to the MCC tree, DuvPO was paraphyletic and not the sister group of the (polyphyletic) Pseudoalveolata. Lastly, in the minimum tree of the PCo2 axis, DuvPO and CylABY were polyphyletic. Meanwhile, Belemnitina and Pseudoalveolata were paraphyletic in the same tree.

In summary, the topology of the two main clusters differs mainly in the position of DuvPO. This clade is more closely related to the Cylindroteuthidae in cluster A but more closely related to the Pseudoalveolata in cluster B. Differences in posterior probabilities are partially caused by different branching positions of DuvPO, breaking up the monophyly of other clades. The minor cluster C splits DuvPO into two clades: one related to the Pseudoalveolata, while the position of the other clade is volatile. Outlier trees contain a relatively high number of aberrant clades that are only present in a tiny fraction of the trees.

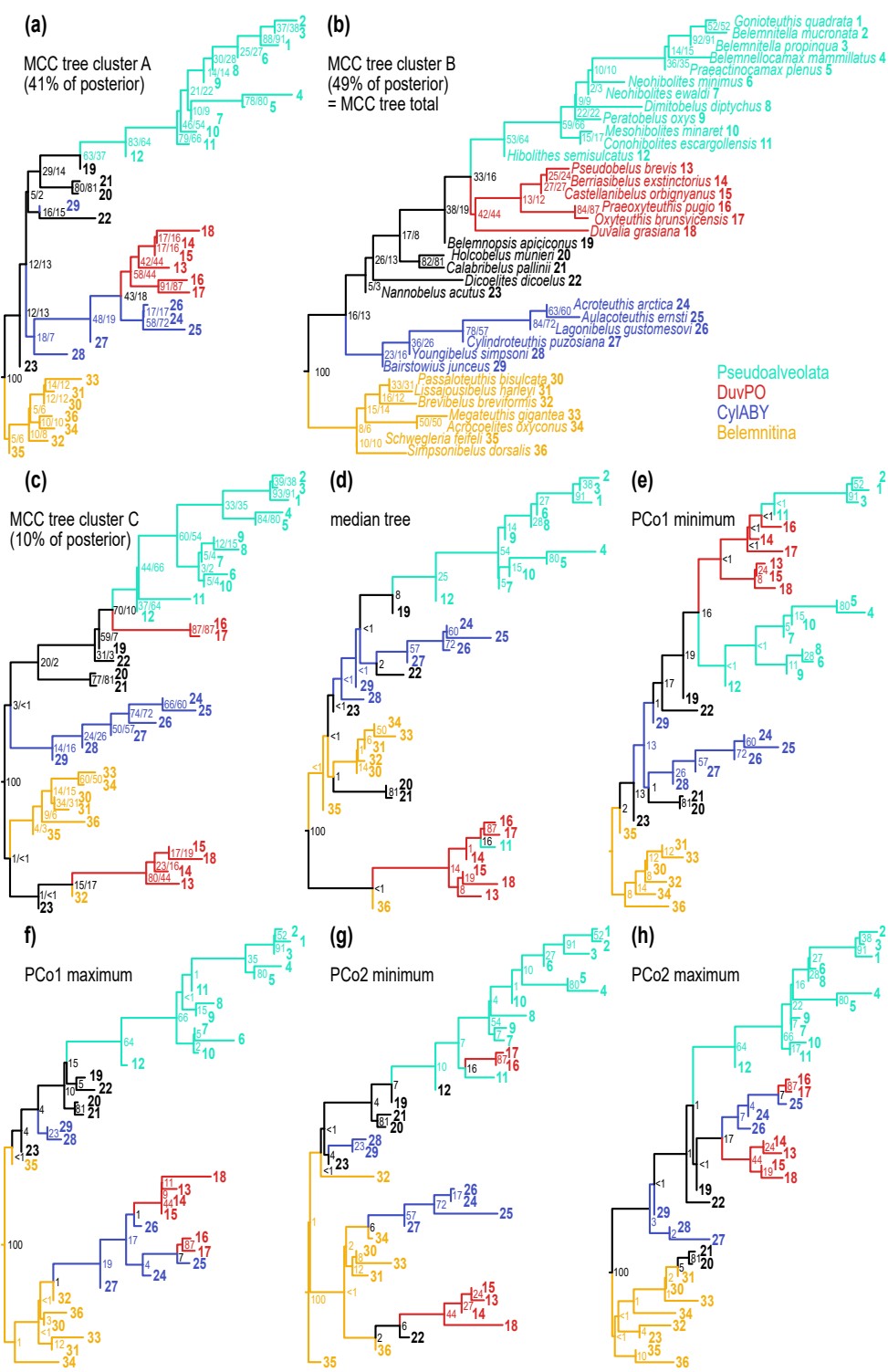

**(a)** MCC tree cluster A (41% of posterior)

**(b)** MCC tree cluster B (49% of posterior) = MCC tree total

Pseudoalveolata
DuvPO
CylABY
Belemnitina

**(c)** MCC tree cluster C (10% of posterior)

**(d)** median tree

**(e)** PCo1 minimum

**(f)** PCo1 maximum

**(g)** PCo2 minimum

**(h)** PCo2 maximum

*Gonioteuthis quadrata* **1**
*Belemnitella mucronata* **2**
*Belemnitella propinqua* **3**
*Belemnellocamax mammillatus* **4**
*Praeactinocamax plenus* **5**
*Neohibolites minimus* **6**
*Neohibolites ewaldi* **7**
*Dimitobelus diptychus* **8**
*Peratobelus oxys* **9**
*Mesohibolites minaret* **10**
*Conohibolites escargollensis* **11**
*Hibolithes semisulcatus* **12**
*Pseudobelus brevis* **13**
*Berriasibelus exstinctorius* **14**
*Castellanibelus orbignyanus* **15**
*Praeoxyteuthis pugio* **16**
*Oxyteuthis brunsvicensis* **17**
*Duvalia grasiana* **18**
*Belemnopsis apiciconus* **19**
*Holcobelus munieri* **20**
*Calabribelus pallinii* **21**
*Dicoelites dicoelus* **22**
*Nannobelus acutus* **23**
*Acroteuthis arctica* **24**
*Aulacoteuthis ernsti* **25**
*Lagonibelus gustomesovi* **26**
*Cylindroteuthis puzosiana* **27**
*Youngibelus simpsoni* **28**
*Bairstowius junceus* **29**
*Passaloteuthis bisulcata* **30**
*Lissajousibelus harleyi* **31**
*Breviibelus breviformis* **32**
*Megateuthis gigantea* **33**
*Acrocoelites oxyconus* **34**
*Schwegleria feifeli* **35**
*Simpsonibelus dorsalis* **36**

**Figure 6.** Sampled trees from the tip-dated analyses. Values at nodes represent proportion of trees that recover the same clade as monophyletic, i.e., the posterior probability of the clade. Where two values are given, the first value represents the proportion of agreeing clades within the same cluster, while the second value refers to the entire dataset. **(a-c)** Maximum clade credibility trees for each of the three tree clusters in the posterior (see Fig. 5). Note that the MCC tree of cluster B is identical to the MCC tree of the entire dataset. Percentages refer to the proportion of trees within the same cluster. **(d)** Median tree, calculated from the entire dataset. This tree represents an average estimate of the phylogeny within tree space. **(e-h)** Extreme topological limits of the first two axes tree space (see Fig. 5). These trees represent opposite outliers of the posterior sample, showing the range of expected topological configurations.

## 3.3 Ancestral states and evolutionary rates

When plotting element ratios as a phenogram on the MCC tree, it becomes obvious that these data are inconsistently distributed across the tree (Fig. 7). For Mg/Ca, there appear to be two distinct groups: (i) one group with consistently low ratios, i.e., between 3.0 mmol/mol (*Acroteuthis*) and 5.3 mmol/mol (*Aulacoteuthis*) and (ii) one with Mg elemental concentrations being higher, between 7.6 mmol/mol (*Neohibolites*) and 16.6 mmol/mol (*Mesohibolites*) (Fig. 7a). This pattern is not only restricted to the average values per genus. Comparing the distributions of each genus reveals the same gap with relatively little overlap between the two groups, although outliers exist in both directions (Fig. 2a).

In the phenogram, Mg/Ca ratios contain rapid changes between neighbouring nodes, e.g., between *Cylindroteuthis* and its sampled ancestor, *Youngibelus*. On this branch, Mg/Ca ratios decrease from 13.3 mmol/mol to 5.1 mmol/mol within approximately 15 Myr (Fig. 7a). The two groups only weakly reflect phylogeny, as the transition from high to low Mg happened at least four times independently. Meanwhile, only a single reversal occurs at the base of the clade comprising Pseudoalveolata and DuvPO. However, note that this observation is based only on the MCC tree and does not account for topological uncertainty.

Conversely, Sr/Ca does not show distinct groups but also generally smaller changes between neighbouring tips (Fig. 7b). Most Sr/Ca are confined to an interval between 1.0 and 2.0 mmol/mol. A few outliers deviate from this pattern with elevated ratios compared to their nearest relatives, such as *Dimitobelus* (2.70 mmol/mol) and *Belemnellocamax* (2.07). A drop from 1.72 to 1.05 mmol/mol in Sr/Ca ratio was found from the sampled ancestor *Youngibelus* and its descendant *Cylindroteuthis*.

Ancestral state reconstructions draw a similar picture (Fig. 8). Mg/Ca ratios of 10.1 mmol/mol are reconstructed at the root (sd = 2.9 mmol/mol). The mean Mg/Ca ratio of all internal nodes was 9.2 mmol/mol and 9.6 mmol/mol when excluding sampled ancestors (sd = 1.9 mmol/mol). The two groups of lower and higher Mg concentrations are visible as well, although except for the node including *Holcobelus* and its sister group, all internal nodes fall into the lower part of the high Mg group, i.e., between 8 and 12 mmol/mol (Fig. 8c). There are two transitions to low ratios (< 6.0 mmol/mol) at the base of larger clades. These are found in the clade comprising the sister group of *Youngibelus* (Cylindroteuthidae) and the sister group of *Nannobelus*. In the latter case, this is followed by a reversal to Mg/Ca ratios of 8.4 mmol/mol (sd = 1.8 mmol/mol) in the sister group of *Belemnopsis* (Pseudoalveolata and DuvPO). The other two transitions to low Mg/Ca ratios were restricted to *Praeoxyteuthis* (5.1 mmol/mol) and *Peratobelus* (4.8 mmol/mol), respectively.

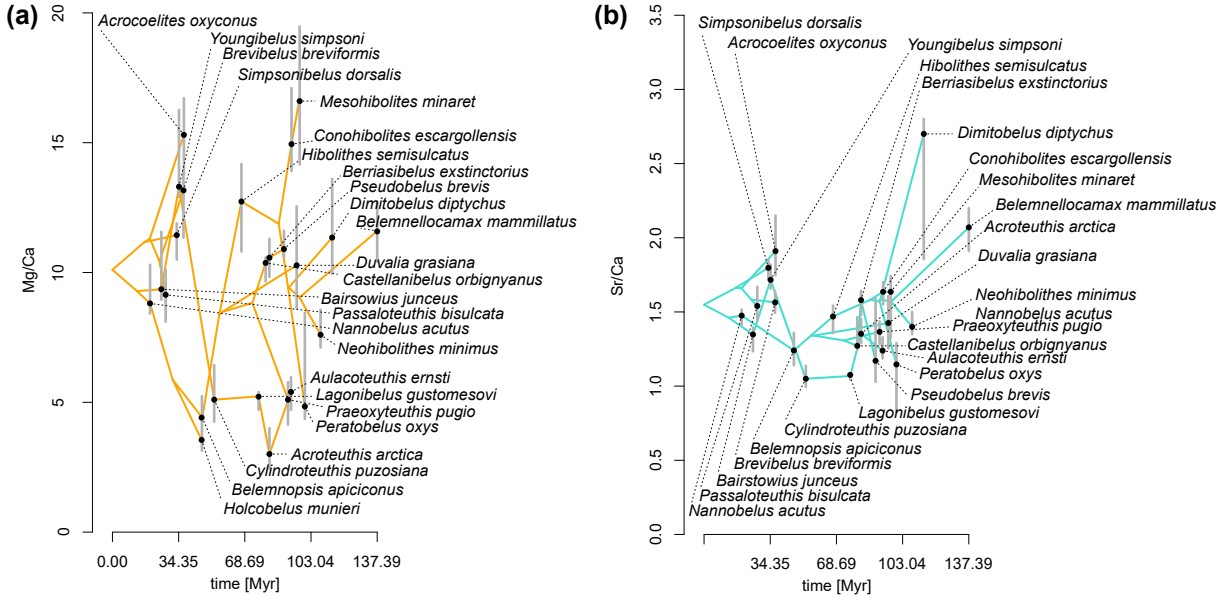

**Figure 7.** Phenogram of element ratios, plotted to MCC tree. Grey bars represent 75% quantiles of the data (cf. boxes in Fig. 2). Note that time is represented in million years since the root of the tree, which is set at 216.3 Ma. The youngest taxon in this tree, *Belemnellocamax mammillatus*, occurs at 78.9 Ma. **(a)** Mg/Ca ratio. **(b)** Sr/Ca ratio.

Ancestral states of Sr/Ca ratios were reconstructed as relatively homogeneous across the majority of the branches with only gradual changes except for the outliers mentioned above (Fig. 8b). The mean Sr/Ca ratio was 1.45 mmol/mol across all nodes (sd= 0.099 mmol/mol) and 1.55 at the root (sd = 0.086 mmol/mol). The ancestral states of Sr/Ca ratios at the nodes fall between 1.05 and 1.72 mmol/mol.

Overall, our simulations of evolutionary rates of Mg/Ca ratios inferred a mean of $\sigma^2$ = 1.70, which means that the standard deviation of change (i.e., the average absolute change, mean change is 0.0 mmol/mol) per 1 Myr is $\sigma$ = 1.30 mmol/mol, and $\sigma$ = 4.12 mmol/mol per 10 Myr. Given that the mean Mg/Ca ratio is 10.51 mmol/mol, these evolutionary rates suggest a change of 12.4 % of the mean value per 1 Myr. The inferred values of $\sigma^2$ for Mg/Ca ratios are consistent regardless of the exemplar tree. The mean of $\sigma^2$ ranges between 0.97 for the MCC tree of cluster B and 2.38 for the median tree (Fig. 9a). Across the

sample of 1000 trees from the posterior, the 95% confidence interval of mean values from all trees was between 0.90 and 3.17 ($\sigma$ between 0.95 mmol/mol and 1.78 mmol/mol per 1 Myr, or 9.0-16.9% of the mean). Notably, the MCC tree of cluster B, which coincides with the overall MCC tree, resulted in a comparatively low mean value of $\sigma^2$ with a narrow confidence interval for both Mg/Ca (Fig. 9a) and Sr/Ca (Fig. 9b).

    The evolutionary rates of Sr/Ca ratios are lower than those of Mg/Ca, but only when given as absolute values. For the entire

sample, we inferred a mean value of $\sigma^2$ = 0.04. In analogy to the above, this corresponds to an average absolute change of $\sigma$ = 0.2 mmol/mol per 1 Myr and $\sigma$ = 0.63 mmol/mol per 10 Myr. The overall mean of Sr/Ca was 1.62 mmol/mol across all rostra in our dataset, i.e., $\sigma$ reflects a change of 12.3 % of the mean per 1 Myr. This percentage is close to the corresponding value

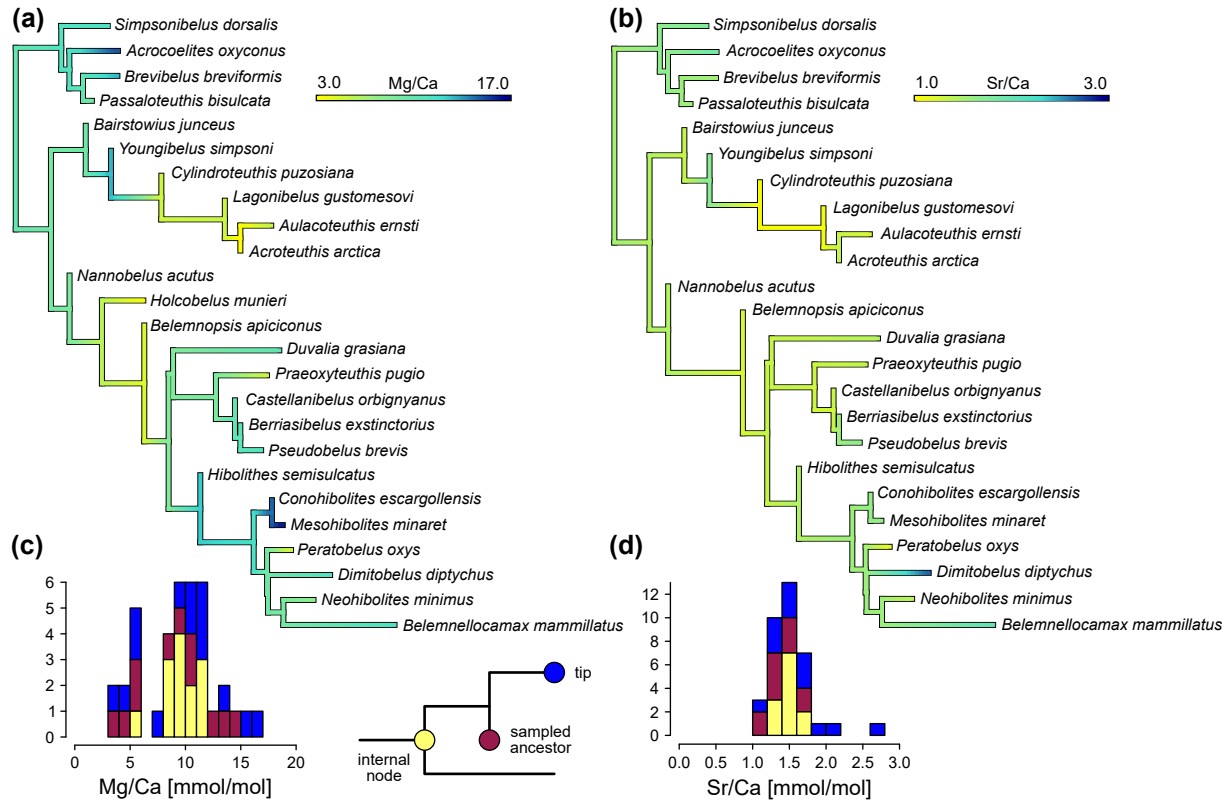

**Figure 8.** Ancestral state reconstructions of elemental ratios, based on MCC tree. **(a)** Mg/Ca ratio. **(b)** Sr/Ca ratio. **(c)** Histogram of Mg/Ca values, coloured by internal node, sampled ancestor and tip. **(d)** Histogram of Sr/Ca values, coloured by internal nodes, sampled ancestors and tips. Note that values at tips and sampled ancestors are directly measured, while internal nodes use average values from ancestral state reconstruction.

for Mg/Ca ratios. Variations in $\sigma^2$ between trees were higher for Sr/Ca than for Mg/Ca, with only slightly or non-overlapping distributions (Fig. 9b). Mean values of $\sigma^2$ in the exemplar trees ranged between 0.010 for the MCC tree of cluster B and 0.628 for the tree with the maximum value on the PCo1 axis. Across the mean values from all posterior samples, the 95% confidence contained values between 0.008 and 0.088 ($\sigma$ between 0.089 mmol/mol and 0.297 mmol/mol per 1 Myr, or 5.5-18.3% of the mean).

When all posterior samples are combined, the distributions of $\sigma^2$ values based on trees from clusters A and B show almost complete congruence, while those inferred from trees of cluster C display a slightly shifted distribution for both Mg/Ca and Sr/Ca ratios (Fig. 10). Note that cluster A and B are roughly similar in size, comprising together about 90% of the trees and cluster C is less dense and occupies an intermediate position in tree space (Fig. 5).

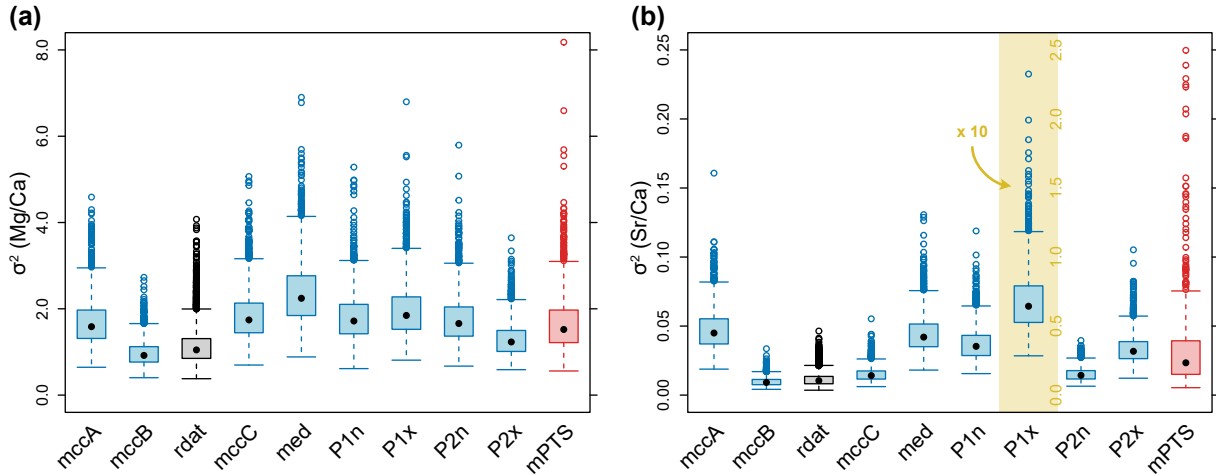

**Figure 9.** Evolutionary rates ($\sigma^2$) of element ratios. Blue boxplots represent posterior distributions inferred from a single tree, while the red boxplot includes the mean of every analysis for each tree from the posterior tree sample (mean posterior tree samples = mPTS). Thus, the mean of each blue boxplot corresponds to a single point of mPTS. The grey boxplot (rdat) corresponds to the analyses of randomly altered data, using the MCC tree of the entire tree sample (= mccB). **(a)** Evolutionary rate of Mg/Ca ratio. **(b)** Evolutionary rate of Sr/Ca ratio. Note that for display reasons, the boxplot in the yellow area (P1x) is rescaled, see corresponding axis labels.

Randomly altered data led to slightly higher values of $\sigma^2$ than the original data for the MCC tree of cluster B (Fig 9, compare mccB vs. rdat). This pattern was observed for both Mg/Ca and Sr/Ca. The variation between the original data and randomly altered data was distinctly smaller than the differences observed using the original data but fixing different trees.

## 4   Discussion

### 4.1   Rostrum phylogeochemistry

Elemental concentrations of belemnite rostrum calcite are influenced by the ambient chemistry of seawater, as well as by kinetic, biological, and post-mortem (diagenetic) factors. Because of the complex interplay of different factors, reconstruction of potential evolutionary constraints (i.e., biological factors) is challenging but crucial for the interpretation of geochemical proxy data derived from these archives (e.g., Ullmann et al., 2015; Immenhauser et al., 2016; Ullmann and Pogge von Strandmann, 2017; Hoffmann and Stevens, 2020, and references therein). Results documented here are in agreement with a taxon-specific distribution of rostrum elemental ratios (McArthur et al., 2007; Wierzbowski and Joachimski, 2009; Li et al., 2013; Stevens et al., 2022). The rostra of *Cylindroteuthis*, *Belemnopsis*, *Acroteuthis* and others show particularly low Mg/Ca ratios (between 3.0 and 5.3 mmol/mol), while those of, e.g., *Acrocoelites*, *Hibolithes* and *Mesohibolites* display higher Mg concentrations (between 7.6 and 16.6 mmol/mol). The distribution of Sr/Ca ratios is comparably homogenous across different taxa and plots into a relatively narrow window between 1.0 and 2.0 mmol/mol. Exceptions include high Sr/Ca ratios in *Dimitobelus* and

*Belemnellocamax*. With regard to Fe/Ca and Mn/Ca ratios, the variability is found to be more significant within the same taxon compared to that between different taxa, which is what would be expected if diagenesis were the primary controlling factor. Another possible interpretation of this pattern is a stronger biomineralization effect, perhaps combined with kinetic factors.

The analyses presented here imply that element ratios in belemnite rostra were subject to high evolutionary rates. The simple Brownian Motion model assumes a constant evolutionary rate across the tree but this is likely a simplification. In particular, short-term environmental disturbances would have probably led to sudden changes in proxy data. Nevertheless, it is important to understand the evolutionary rate as an increase in variance through time, and not as projected absolute changes in the trait value (i.e., element/Ca). Specifically, under our model, the expected average change for element/Ca within 1 Myr is 0.0,
meaning that positive and negative changes cancel each other out. The expected absolute average change is $\sigma = 1.30$, which means that there would be lineages expected with stasis or minor changes, as well as lineages with larger changes. Thus, the evolutionary rate estimates should not be taken literally for individual lineages but understood in the context of the model specifications as a global average of a stochastic process. We also stress that $\sigma^2$ is simply a metric for the tempo of changes in the trait value, independently of the underlying causes.

As for the primary data, differences in element ratios may be, in part, biased by differences in the seawater properties inhabited by the belemnites or inherent variability in the form of variable degrees of diagenetic alteration overprinting the primary geochemical composition of the rostra.

Nevertheless, our sensitivity tests with randomly altered data suggest that the estimation of evolutionary rates is robust against the inclusion of some samples that may be diagenetically altered. Especially in taxa with a high number of samples, it
would take a lot of data points to significantly affect the median of the Mg/Ca or Sr/Ca ratios. In comparison, uncertainty in tree topology had a larger effect on $\sigma^2$ (Fig. 9). As the values for $\sigma^2$ reported above account for uncertainty in tree topology, we consider these estimates to be reliable despite some inherent potential for biased data. The inclusion of geochemical data from further taxa would likely refine the estimates. As the estimated rates appear to be independent of topological uncertainty to some degree, more precise estimates may be achieved even if phylogenetic relationships are not always known in detail.

Seawater properties (water depth, latitudinal differences, basin-specific properties of aquafacies etc.) influenced the composition of the body fluids of belemnites, from which the rostrum was precipitated. Thus, these environmental properties are thought to have a direct impact on the chemical signature of the rostrum (Gröcke and Gillikin, 2008; Immenhauser et al., 2016). Even though these represent non-genetic and, therefore, phylogeny-independent factors, we argue that reconstructing evolutionary rates can still be meaningful. This is because it seems unlikely that the geochemical proxy data of different taxa would
record sudden and non-systematic shifts in seawater properties, with the possible exception of major short-term disturbances (e.g., the Toarcian Oceanic Anoxic Event, see Ullmann et al., 2014). Moreover, due to the stratigraphically and geographically restricted ranges of belemnite taxa, a single species would not be expected to record significantly differing seawater properties. Widely distributed generalist and long-ranging taxa are expected to display a broader distribution in elemental ratios. Evolutionary rates would arguably reflect the frequency of habitat shifts (coastal versus open oceanic settings). Nevertheless, as
differences in element ratios of belemnite taxa from the same locality exist (Fig. 4), seawater properties are probably not the only contributing factor.

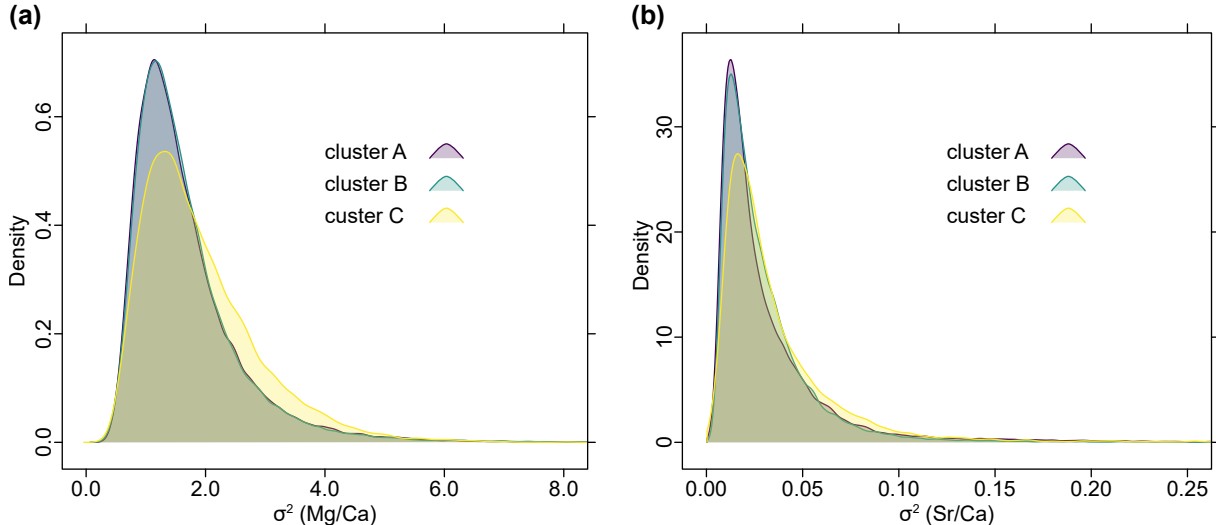

**Figure 10.** Density plots of combined posterior samples of $\sigma^2$ inferred from all trees, grouped after tree clusters (see Fig. 5). Note that cluster A and cluster B almost entirely overlap, while cluster C is slightly shifted. **(a)** Distribution of evolutionary rate of Mg/Ca. **(b)** Distribution of evolutionary rate of Sr/Ca.

Post-mortem (diagenetic) alteration and poorly constrained kinetic (disequilibrium) effects are significant problems in the analysis of evolutionary rates. That said, it seems likely that the proxy data analysed should display some degree of homogenisation if diagenetic resetting were a dominant control. Diagenetic homogenisation would be most pronounced for specimens

in a given section that has seen above-average diagenetic overprint (i.e., deep burial and hydrothermal or meteoric alteration, etc.), independent of taxonomy.

The ancestral state reconstructions and inferred evolutionary rates suggest that element ratios show a complex pattern with frequent clade-independent changes despite their taxon-specific distributions. There are several, not mutually exclusive explanations for this pattern:

*1) High evolutionary rates are a genuine feature of belemnite rostrum geochemistry*. Given that the dataset used includes a limited amount of taxa, the changes are perhaps too rapid to be accurately detected. A much denser species sampling would be required to support or reject this hypothesis. Given that the amount of available characters is limited, producing a phylogeny with the required level of accuracy is challenging. In addition, for many species, only a limited number of well-constrained elemental concentrations are available in the literature.

Moreover, the identification of species in published works is often difficult to confirm. This is because sample extraction (usually drilling of powder samples or crushing of bulk rostra) is a destructive process. Not every specimen is documented both before and after sampling. Moreover, the identification at the species level was not necessarily the aim of the published studies, many of which focussed on palaeoceanographic aspects and often lacked guidance by belemnite taxonomists. This effect is

demonstrated by the fact that taxa in open nomenclature (i.e., only identified to genus level) have relatively wide ranges of
element ratios (Fig. 3). This indicates that the specimens may belong to different species.

A wide range of Mg/Ca data for persistent and widespread taxa like *Hibolithes* might also be caused by hidden genetic
diversity that is not observable in rostrum morphology alone. It always has to be kept in mind that the rostrum represents a
phylogenetically informative character, but every 'rostrotaxon' probably contains several 'biological taxa' and potentially vice
versa. This can be compared to the situation in sepiids, in which the congruence of taxa (genera/subgenera/species) based
on genetic markers and cuttlebone morphology is complex (e.g., Lupše et al., 2023). A better understanding of the chemical
signatures of different species is urgently needed for geochemical studies but may also have some taxonomic relevance. We
thus encourage the inclusion of geochemical data in taxonomic studies, if available. If homologous spatial and ontogenetic
areas of the rostrum are targeted, it may be possible to identify 'chemospecies' within a contemporary assemblage.

*2) The elemental concentrations are strongly ontogenetically and spatially controlled.* Differences in proxy data related to
ontogenetic stages and position within the rostrum are not always reported. That said, there is evidence that element ratios of
rostra vary during ontogeny and a trend towards higher Mg/Ca ratios in early ontogenetic stages and close to the apical line
of several taxa was reported (Ullmann et al., 2015; Stevens et al., 2017, 2022; Ullmann and Pogge von Strandmann, 2017).
Depending on the volume of the subsamples drilled, later ontogenetic stages and parts that are further away from the apical
line might be overrepresented in our data as these form larger portions of the rostrum and, hence, are more likely sampled or
dominate the signature of bulk sampled rostra. Examples of sampling that include highly resolved ontogenetic and spatial data
are *Passaloteuthis* (Ullmann et al., 2015), *Neohibolites* (Stevens et al., 2017) and *Duvalia* (Stevens et al., 2022), all of which
contain outliers in Mg/Ca and to a lesser degree in Sr/Ca. However, the bulk of the data overlaps with data from other genera
that were not sampled with a focus on ontogenetic stages (Fig. 2a).

*3) Patterns in rostrum proxy data reflect genuine patterns in seawater hydrogeochemistry* (see Hoffmann and Stevens, 2020
for discussion and critique). If this holds, element concentrations, and hence ratios, are expectedly controlled by phenotypic
plasticity as opposed to evolutionary (i.e., genetic) constraints. Detecting plasticity in the fossil record is challenging because
genetic and environmental factors are not easily separated (Lister, 2021). Recent coleoid cephalopods are known for increased
levels of plasticity (e.g., Boyle and von Boletzky, 1996), which may, in part, be due to extensive amounts of RNA editing
(Liscovitch-Brauer et al., 2017). It may also be possible that each belemnite taxon has a distinct partition coefficient for each
element, which may be used to calculate seawater properties. However, this view is likely too simple, as complex pathways
form mollusc biominerals, and bodily fluids differ in their elemental composition from seawater (Immenhauser et al., 2016).
Thus, calculating such coefficients would be non-trivial and difficult to test since the original seawater composition is unknown.

*4) The heterogeneous dataset includes a wide range of taxa.* This heterogeneity might include a sampling and analytical
bias. Even if most studies use the same analytical method (e.g., wet chemical analysis), minor technical differences such as
experimental protocols, equipment or even human factors might play a small role. In the case of sample material that was
drilled mechanically from rostra, the position of the sampling point and the volume of subsamples might affect the resulting
data (see, e.g., Ullmann et al., 2015). If all data were collected for a single study, good scientific practice would dictate that the
same protocol would be used for every sample. For obvious reasons, this was not possible for our study. All of these features

are difficult to constrain, but it is argued that the effect is likely negligible but, if present, should be non-systematic (i.e., not biased in a particular direction, but at most produce more noisy data).

Besides the phylogenetic patterns, we noted that there is a strong correlation between the evolutionary rate $\sigma^2$ of the Mg/Ca and Sr/Ca ratios (Spearman's rank correlation: $\rho = 0.57$, p-value < 0.001; Fig. 11b). It seems likely that this feature is caused by the general correlation of Mg/Ca and Sr/Ca ratios across all belemnite rostra in our dataset (Spearman's rank correlation: $\rho = 0.49$, p-value < 0.001; Fig. 11a). Alternatively, it might also be argued that if element ratios (of individual specimens) would be distributed independently of phylogeny and taxonomy, a direct link with evolutionary rates (the average of individual trees) is not necessarily to be expected. At the level of a working hypothesis, we take this notion as evidence for a taxon-specific vital effect of element ratios. Another possibility is that this pattern is caused by extreme outliers being relatively short-lived – since Mg/Ca and Sr/Ca are correlated, a peak would decrease more rapidly in both, while lower element ratios would remain stable for a longer period.

Our study shows that rostrum Mg/Ca ratios if properly understood, need to be interpreted in an evolutionary context. Cylindroteuthids appear to contain relatively low Mg/Ca ratios, while the Pseudoalveolata and Belemnitina have relatively high Mg/Ca ratios. However, these are general trends, and the changes are rapid in some cases. There is a strong overlap between individual taxa, suggesting that identifications based on element ratios alone are not possible. Thus, element ratios should be used carefully as a character in phylogenetic analyses. It also shows the importance of identifying belemnite material as accurately as possible before using it for geochemical studies.

In contrast, they are well suited for the reconstruction of ancestral states and inference of evolutionary rates. Future studies may also investigate their potential in species-level analyses, where rapid changes can be more easily detected, and the amount of characters is particularly poor. In contrast, Sr/Ca ratios appear to be relatively independent of phylogeny, although they still show a more or less taxon-specific distribution. Comparing the two elemental ratios implies that there was some genetic control on the incorporation of Mg into the rostrum. Presumably, this is related to the biomineralization strategy that underwent evolutionary changes between taxa. Alternatively, the discrepancy might be related, at least in part, to solid solution geochemistry; specifically, the ionic radii of Mg and Sr. Mg ionic radii (0.72 Å) are more suitable to substitute Ca (1.0 Å) in the calcite crystal lattice. In contrast, Sr ionic radii (1.31 Å) are more suitable for substituting Ca in the aragonite crystal lattice (see Immenhauser et al., 2016 for details and explanation). For the calcitic belemnite rostra, this means that (arguably) the amount of Mg was more variable than Sr, requiring fewer genetic modifications in the biomineralization apparatus.

In conclusion, rostrum element ratios likely represent a complex interplay of evolutionary, ontogenetic, environmental, kinetic and diagenetic factors, which all need to be accounted for if these archives are to be used as geochemical proxies. Diagenetic bias can be reduced by various screening processes, including cathodoluminescence analysis, although there are potential pitfalls (see Stevens et al., 2022). Ontogenetic variation can be accounted for by appropriate sampling strategies and either compare their trajectories or only compare similar ontogenetic stages (e.g., Ullmann et al., 2013, 2015; Sørensen et al., 2015; Stevens et al., 2017, 2022; Ullmann and Pogge von Strandmann, 2017). The sampling position within the rostrum needs to be specifically tailored to align with the research goal, i.e., a study on environmental effects will likely require a broader sampling. In contrast, for phylogenetic studies, sampling the same position and the same ontogenetic stage is important due

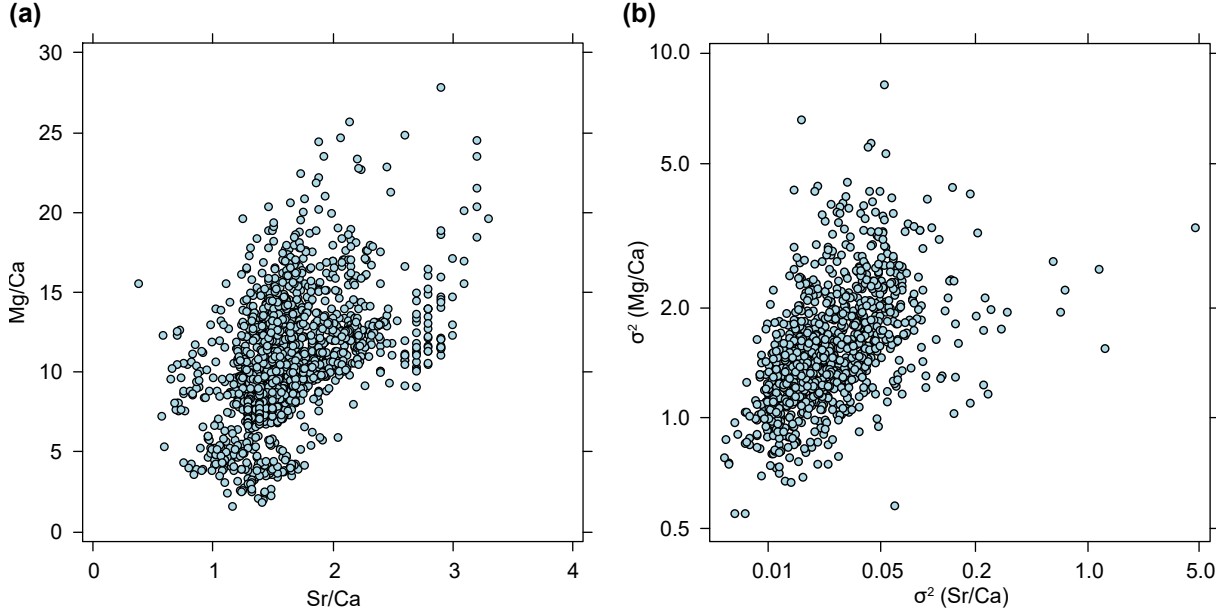

**Figure 11.** Correlation between element ratios and their evolutionary rates. **(a)** Element ratios as raw values, taken from entire data set, i.e., each point represents a single measurement from a belemnite rostrum (n = 2241). **(b)** Evolutionary rates ($\sigma^2$), taken as means of posterior samples, i.e., each point represents the average evolutionary rate inferred for a single tree (n = 1000). Note that B is shown on a double logarithmic scale.

to homology. To account for evolutionary constraints, the taxonomy of the studied specimens should be precisely controlled, ideally with well-documented reference images and/or measurements. This complexity calls for a greater interdisciplinary effort to collaborate between geochemists, oceanographers and palaeontologists when interpreting proxy data from belemnite rostra.

## 4.2 Belemnite phylogeny

Reconstructing a well-supported phylogeny was not a primary goal of this study, though extending previous results from Stevens et al. (2023) was a necessary first step. Consequently, our results provide additional insights into this still largely uncharted area. We find further support for the Pseudoalveolata (Stevens et al., 2023), while the position of the Duvaliidae remains obscure, with some support for its closer association with the Pseudoalveolata. The decreased support of the Pseudoalveolata in comparison with Stevens et al. (2023) is partly caused by the uncertainty in the position of the DuvPO clade, which was occasionally recovered as forming part of the Pseudoalveolata. This question requires further investigation, e.g., by analysing the rostrum microstructures of additional species. Improved species sampling may also be useful, as the number of Jurassic and Early Cretaceous pseudoalveolates and duvaliids is still rather low in our matrix.

In comparison with the earlier study, our analysis also included a denser sampling of Duvaliidae, Cylindroteuthidae and Oxyteuthidae. These three families previously formed a (poorly supported) monophyletic clade together with *Dicoelites* (Stevens et al., 2023) but were here resolved differently. Instead of being a sister group to the Cylindroteuthidae, our results suggest that *Oxyteuthis* and *Praeoxyteuthis* are more closely related to the (paraphyletic) Duvaliidae, while *Aulacoteuthis* received high support as a member of the Cylindroteuthidae. Mutterlose and Baraboshkin (2003) suggested that the Hauterivian boreal *Aulacoteuthis absolutiformis* belongs to the Cylindroteuthidae, while the mid-Barremian *Aulacoteuthis* from northwestern Europe (which is included here with *A. ernsti*) was thought to be a member of the endemic Oxyteuthidae. Our results imply a possible alternative, with the cylindroteuthid *Aulacoteuthis* migrating into Europe during the mid-Barremian, contrasting with the conclusion of Mutterlose (1983) based on a biometric, stratophenetic approach.

The previous analysis suggested a monophyletic Belemnitina that excludes the families Cylindroteuthidae and Oxyteuthidae (Stevens et al., 2023). Our results suggest alternative scenarios where the 'Belemnitina' represent an ancestral paraphyletic group from which later belemnitid lineages evolved. However, topological uncertainties are particularly high near the base of the tree. Note that we excluded sinobelemnitids here, which may be relevant for the early diversification of the group but are still poorly known (Zhu and Bian, 1984; Iba et al., 2012; Ma et al., 2023). Neither microstructures nor geochemistry of sinobelemnitids have been investigated so far.

The volatility of belemnitid relationships retrieved from this study and compared to the previous analysis highlights that further research is needed. The number of characters needs to be expanded in the future, e.g., by implementing geometric morphometric data (e.g., Dera et al., 2016). Further insights may also be gained by including taxa from several currently un(der)sampled families (e.g., Hastidae, Salpingoteuthidae).

Beyond belemnitid relationships, our study serves as a practical example of how topological uncertainties within a posterior tree sample may be investigated using available tools of tree space analysis. In palaeontological phylogenetic studies, only a single reference tree is usually given with support values reflecting uncertainty, e.g., posterior probability in trees inferred with Bayesian methods or bootstrap support in Parsimony trees (e.g., O'Reilly and Donoghue, 2018). Summary trees, however, represent only a point estimate, and clade supports are difficult to interpret because alternative topological configurations are not obvious. By using multidimensional scaling and analysing individual tree clusters (Jombart et al., 2017), we show how to investigate the extent and potential sources of topological incongruence in trees resulting from a single analysis. In particular, we suggest that using partial posterior probabilities of clades only within clusters of trees is helpful in identifying clades with conflicting positions between tree clusters. For example, a clade may show weak overall support but highly supported, incompatible positions in different tree clusters. The approach thus allows us to explicitly formulate alternative phylogenetic configurations rather than generally reporting high topological uncertainty. Furthermore, by reporting trees with maximum and minimum values on the principal coordinate axes, it is possible to assess the limits of the topological variation within the posterior tree sample.

### 4.3 Implications for chemical diagenesis-screening of belemnite rostra

Screening of belemnite rostra calcite for evidence of post-mortem alteration (note, diagenetic modification of biominerals can commence during the lifetime of an organism, Immenhauser et al., 2016) has been a topic for many decades. Besides optical methods such as SEM imaging and cathodoluminescence (e.g., Saelen, 1989; Barbin et al., 1991), element concentrations were introduced as a sensitive tool to assess alteration (Veizer, 1974). The increase of Mn and Fe elemental concentrations and the decrease of Sr and Mg concentrations are thought to indicate diagenetic alteration (e.g., Veizer, 1983; Ullmann and

Korte, 2015). The threshold limits were set at variable values, depending on the local context (cf. Dutton et al., 2007; Price and Mutterlose, 2004; Wierzbowski and Joachimski, 2007). Moreover, recent work has documented that diagenesis may result in fabric-retentive carbonates with altered geochemical proxy values or, *vice versa*, in fabric-destructive carbonates with largely preserved marine geochemical signatures (Bernard et al., 2017; Immenhauser, 2022; Mueller et al., 2024).

The results shown here are in agreement with the work by Stevens et al. (2022), suggesting that the threshold limits for Mn

and Fe elemental concentrations have usually been set too high in many earlier studies. Due to potential environmental (different water masses with different geochemical properties through which the nektic belemnites move), kinetic (non-equilibrium fractionation) and biological (biomineralization strategies) impact on Mg/Ca and Sr/Ca ratios, we question the validity of specific threshold limits on screening approaches. We tentatively propose that the values indicated in Table 4 might give hints for more refined threshold limits for Mn/Ca and Fe/Ca ratios. Additionally, Fe/Ca and Mn/Ca ratios may also be compared at

the genus level (Fig. 2c, d). However, we highlight that in state-of-the-art diagenesis screening of fossil biominerals, screening techniques should not be used in isolation, and chemical assessments need to be combined with thorough analyses by microstructural, SEM-based methods or cathodoluminescence microscopy.

### 5 Conclusions

'Phylogeochemistry' is here proposed as a new term for the application of Bayesian phylogenetic tools to study evolutionary

pathways of the chemical composition of fossil biomineralising organisms. Here, we tested the utility of element/Ca ratio data as a phylogenetic character. Our results show that evolutionary rates of Mg/Ca and Sr/Ca ratios in belemnites are relatively high, so they are likely of limited use in reconstructing evolutionary relationships unless applied at a high taxonomic resolution. The heterogeneous changes in Mg/Ca provide insights into the evolution of biomineralization patterns. Our study highlights that care must be taken when interpreting geochemical data of belemnite rostra, as evolutionary constraints cause biased

element ratios, particularly if samples are taken from distantly related taxa. Therefore, it is crucial to accurately document the taxonomic identity of the analysed material in geochemical studies. For future research, we propose detailed analyses on a lower taxonomic level to analyse short-term changes in element compositions.

Phylogeochemistry has a high potential for future studies, as using this approach makes it possible to link proxy data with taxa in a time-calibrated phylogeny directly. Thus, rather than comparing a phylogenetic tree to a (global) curve of proxy data,

each tip in the phylogeny will have associated proxy data, ideally reflecting local environmental conditions. This opens the door for a large amount of potential applications by adding an evolutionary dimension. Potential applications are, e.g., testing the

influence of local temperatures on extinction or speciation rates or the frequency of evolutionary transitions between cold water and warm water clades. Naturally, this kind of study requires either strongly environmentally-driven proxy data or a profound understanding of their evolutionary constraints. Performing comparable studies for isotope data ($\delta^{13}$C, $\delta^{18}$O) is, therefore, an obvious next step. Our methodology can also easily be transferred to other groups of fossil or even living biomineralising organisms. Last but not least, the flexibility of Bayesian techniques allows for extending the complexity of the models, e.g., by incorporating locality-specific factors to account for environmental effects or testing alternative evolutionary models that allow for variable rates (relaxed Brownian Motion) or evolutionary optima (Ornstein-Uhlenbeck). This study thus represents an important starting point for an evolutionary perspective on geochemical archive data and offers a plethora of opportunities for further research.

*Code and data availability.* All script and data files are available in the supplementary material at https://doi.org/10.5281/zenodo.14004248 (Pohle et al., 2024). File S1 includes the literature dataset of element ratios of belemnite rostra (including calculated missing values). S2 is a zip file that contains the RevBayes scripts files used for the phylogenetic analyses and estimation of evolutionary rates, the nexus file containing the morphological character matrix and the output log and trees files. Mg/Ca and Sr/Ca ratios for all genera and estimated values at nodes are listed in file S3.

*Author contributions.* KS, AP and RH conceptualised the study; AP assembled literature data with the help of KS and RH, developed the methods, wrote the script files, analysed the data and produced the figures; KS and AI acquired funding; AP wrote the initial manuscript with input from KS, RH and AI; all authors revised and edited the manuscript.

*Competing interests.* The authors declare that they have no conflict of interest.

*Acknowledgements.* We thank Laura Jonas for her help with calculations of some of the compiled element ratios. We are grateful for the reviews by C. V. Ullmann and M. Vickers, who greatly helped to improve the final version of the manuscript. This study was funded by DFG project no. 507867999.

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
