# Peer review of "Phylogeochemistry: exploring evolutionary constraints on belemnite rostrum element composition"

_EGUsphere, 2024_

## Referee Comment (RC2)

    tologica Sinica, 23, 300–317, 1984.

[referee-annotated manuscript omitted]

---

## Author Response (AR1)

**Thank you for the opportunity to submit our revised manuscript for publication in Biogeosciences. We revised the manuscript according to the suggestions by the reviewers. We accordingly implemented our proposed changes, see the discussion of the manuscript on egusphere. In addition, we list our responses to the editorial decision in bold. According to these changes, the Figures 2, 7 and 9 have been slightly modified.**

Thank you for posting your author replies to the two review reports. Both reviewers are overall supportive of your work, but they also flag concerns which should be addressed in a revision. After reading through the review reports and your replies, I believe that this manuscript can become acceptable after major revisions. Especially Reviewer 1 raises some critical points, many of which you have replied to satisfactorily and some of which rest on some confusion regarding the methodology. Regarding the latter, I believe the manuscript would likely benefit from some clarification of the methodology and an attempt to make it more accessible for a somewhat broader audience (as per the comments and replies to the report of Reviewer 1).

**We have clarified the methodology, with explanatory comments what the model parameters represent. Importantly, we realised that part of the confusion of Reviewer 1 was likely caused by a missing paragraph and a title of a subsection of the methodology, which somehow got lost during the copying process from a Word version to a LaTeX version. Unfortunately, this missing part covered critical information regarding the morphological data used in the study. Furthermore, this error led to the integration of the rest of the morphology-based phylogeny section into the section on the geochemical data collection. We added the missing part and apologise for this small but consequential mistake. We made sure that the revised version is complete.**

In addition to the changes suggested, I would suggest that you carefully consider the following aspects in your revision:

In their first point, Reviewer 1 poses the question whether the data underlying the study are sufficient to draw conclusions about the taxonomy. I think this question was wrongly interpreted in the Author Comment: The reviewer does not actually ask for more (new) data to be added to the study. Instead, if I read the question correctly, they would like to see a demonstration that the structure of the dataset (with a large part of the data coming from a small number of specimens) does not negatively affect the conclusions concerning the phylogeny. It seems that this could be achieved by doing a statistical test, and the authors should consider this. In its simplest form, the authors could average (or take a median value of) all the data per taxon before the analysis to check if that gives the same outcome in terms of clearly separate chemical compositions for different taxa. I believe a test like this would be a good addition to the manuscript.

**As explained in our responses, we did not draw any conclusions about the taxonomy based on geochemical data (this should now be clear, see previous point). The proposed statistical test (average of all data per taxon) is essentially already present in the manuscript, see Figure 2, where the distributions of all element ratios for each taxon are shown. It would be possible to use a statistical test (e.g., Kruskal-Wallis test) but the separation between different taxa is already so clearly visible that we do not see an added benefit here. Nevertheless, if the editorial team insists on such a test, we are happy to include it in a final version.**

Similarly, in reply to comment 5 of Reviewer 1, it would be useful to add a "sensitivity test" to demonstrate that including the potential diagenetic data from Sørensen et al. does not significantly affect the conclusions, as stated by the authors in their reply.

**Thank you for this suggestion. Since the potential diagenetic data from Sørensen et al. make up less than 5% of the total dataset and affect only a single taxon, we anticipate that this would change very little in the overall result. Instead, we performed a sensitivity test by randomly altering the input values (i.e., element ratios) of all taxa, on average by about 10% (see updated Methodology for details), and repeated this process 100 times. Note that this is a change to the median value, so depending on the sample size of the taxon, it would take quite a lot of data in reality. This would represent a hypothetical, severely biased dataset compared to a slight change in a single taxon. Nevertheless, and to our surprise, the results appear to be quite robust against such alterations, which strengthens our conclusions.**

In reply to the third main comment by Reviewer 1, I agree with the authors that it is a good idea to highlight the applications of the presented approach a bit more clearly in the discussion. In addition, the authors may consider reviewing the methodology section to make sure that their approach for linking chemical data to phylogeny is clear, as the reviewer was apparently confused by this.

**We considerably extended the discussion to outline potential applications. As for the second part of the comment, see above.**

Re the comment of Reviewer 1 to line 166, I would agree with the reviewer that the median value would probably be a more meaningful statistic for describing these data. If it is not too much effort, I would suggest using the medians as descriptive statistics.

**We checked the values again and noticed that the mean value in the manuscript was actually the result of a typo. Thus, we had used the median value all along,**

**presumably for the same reasons. Apologies for this oversight – it was calculated at a very early stage of the study and then apparently incorrectly noted.**

Finally, please carefully consider the general point by Reviewer 2. It seems that a more detailed discussion of the value of Phylogeochemistry including recommendations for future studies to flesh out this new research niche are warranted.

**This was also highlighted by Reviewer 1, so the discussion is now considerably extended and we provide examples and recommendations for future studies.**

---

## Author Response (AR2)

Dear Editor Niels de Winter,

thank you for the detailed reviewer reports and the chance to improve our manuscript. Below, we provide point-by-point responses to the reviewer's and Editor's comments in red.

**Editor:**

Reviewer #2 is quite critical about your study design (especially your choice of working with belemnite data), but concludes that the innovative idea of "Phylogeochemistry" and the approach you took (albeit perhaps more easily applied on other taxa) merits publication as a proof of concept. I would agree with that assessment and therefore would like to give you a second opportunity to revise your manuscript to better reflect that aim and acknowledge the caveats of using belemnite chemistry for this purpose.

We added a few sentences to the discussion. However, despite all the shortcomings, we think that it would be difficult to find taxa to which our approach could be applied more easily. Essentially, three conditions need to apply to a group in order to use our methods: i) a well-resolved, time-scaled phylogeny, and ii) geochemical proxy data with dense taxonomic sampling. Condition i) is fulfilled mainly in vertebrates, which often lack taxonomic coverage in proxy data. While phylogenies are widely available for living taxa due to molecular studies, these phylogenies mostly exclude fossil taxa and therefore, the reconstruction of ancestral states in deep nodes would be particularly uncertain. As for condition ii), many of the main biogenic carbonate archives (e.g., brachiopods, foraminifers) lack quantitative phylogenetic studies, so our approach is not applicable here. It thus becomes clear that a much wider sampling and further taxonomic and phylogenetic work is required in this regard. It may be that other proxies would have provided more interesting insights than element/Ca ratios in belemnites, but this is impossible to know in advance.

We would also like to point out that we were happy to see another recent study published in Biogeosciences that came to similar conclusions by focussing more on the geochemical side and in a different taxon (Boscolo-Galazzo et al. 2025). Accordingly, we included this study briefly in our introduction and discussion.

**Reviewer #1:**

For the revised version of the study "Phylogeochemistry: exploring evolutionary constraints on belemnite rostrum element composition" by Alexander Pohle and co-authors the documents were updated based on reviewer comments and a comprehensive point-by-point response to the comments was provided.

Following the revisions I find that the methodology is now much clearer, with the approach to reconstructing a phylogeny based on morphological characters that then forms the background for interpretation of geochemical variables explained in detail.

Thank you

A few points remain that I think would benefit from a bit more work:

Use of data:

It would be quite useful in my opinion if the authors could make it clearer in their supplementary data appendix which data were used for their phylogeochemical experiment. It is stated in L118 that only taxa with 10+ specimens were considered for the work, and this is repeated in L239. However, as far as I can tell only data for three specimens of *Belemnellocamax* are listed in the appendix (despite a large number of measurements for each individual), yet this genus is incorporated in the discussion? Perhaps this is the only exception and this may be acceptable given the number of measurements available for the few specimens, but nevertheless I think it should be completely clear what data was used.

Apologies, this should read "measurements" instead of specimens and was corrected accordingly. We also highlighted the data in the supplement that were used in subsequent analyses.

There is also a distinction to be made between the "complete" dataset, which also includes taxa with, e.g., uncertain identifications, and the "cleaned" dataset, which was used for the phylogenetic comparative analyses. For all analyses, we used the reduced dataset, but we provide the complete dataset in the supplement for convenience. This should now be clear from the description in the manuscript and the highlighted entries in the supplement.

It would also be worth to critically check the supplementary data for consistency and accuracy. I noted the statement in L264 regarding the unusually low reported Mg/Ca in *Passaloteuthis* for the Pliensbachian of Ukraine. These very low values in all likelihood arise from an erroneous conversion into Mg/Ca ratios analogues to the mistaken conversion of Sr concentrations to Sr/Ca ratios in the cited study. These can be corrected by multiplying the listed Mg/Ca ratios by [M(Ca)*M(Ca)]/[M(Mg)*M(Mg)]. Note that the authors assumed M(Ca) = 40 g/mol and M(Sr) = 87 g/mol. It might be that the authors assumed M(Mg) = 24 g/mol rather than 24.305 g/mol. A similar issue may also be expected for Arabas (2016), for which the supplementary data table looks very similar, but no Sr/Ca ratios are listed, which makes it harder to evaluate this.

We checked the supplementary material again for consistency and accuracy.

Regarding potential conversion errors: Already in the first version, we had double checked all supplied element/Ca ratios by providing the ratio given in the original publication (columns Q, S, U and W in the supplementary table), in addition to the calculated ratios using our formula (columns R, T, V and X; note that if no raw element concentrations were given in the study, this column corresponds to the previous columns). This revealed a few cases where element/Ca ratios were apparently miscalculated, e.g., Sr/Ca ratios mentioned by the reviewer for Arabas et al. (2017). In such cases, we used our calculated values, since we deem the raw values of the element concentrations to be more reliable. We added a statement to make this clear and also report which studies were affected by this.

Since it was not clear to us before what caused this conversion error in Arabas et al. (2017), we are grateful to the reviewer for pointing out the likely source. Unfortunately, neither raw Mg concentrations nor Fe/Ca nor Mn/Ca ratios were supplied in the supplement of Arabas et al. (2017), meaning that it is impossible to tell whether Mg/Ca, Fe/Ca or Mn/Ca ratios were similarly miscalculated as Sr/Ca. In Arabas (2016), there are no raw element concentrations listed either. Even though these values appear suspicious, the evidence is ambiguous and could only be corrected by getting the original data from the authors or a new study on this locality. We feel that it is out of the scope of our study to correct previously published data, but we added a corresponding comment in the revised manuscript that this may be a reason for the lower values. As this affects *Passaloteuthis*, for which we have a large data set (not the least thanks to the reviewer's extensive work), excluding these values would only minimally change the median value for this genus.

During our check of the supplement, we noticed that the "10 measurements per genus" was an error from an earlier version – it should actually mean 5 measurements. Although this sounds admittedly low, it is an unfortunate consequence of the rather poor taxonomic reporting of previous studies, which we already criticised in the discussion. Statistically, it does not make a big difference (i.e., both are uncertain), but it allowed the inclusion of three additional taxa, which is why we made this decision. We would also like to point out the sensitivity analyses (see further details below), which support that a few datapoints with incorrect data would not have a big impact on the results. Moreover, we already conclude that the geochemical data show no clear phylogenetic pattern, and it seems to us that it would be very unlikely that we would suddenly observe a pattern that correlates with the phylogeny, and it seems that this is also not what the reviewer would expect. The question is, therefore, what additional results/insights could be gained from a "perfect" dataset, the latter of which is difficult to achieve.

While we appreciate the concerns of the reviewer for data quality, we would like to point out once more that the dataset is inevitably heterogeneous, and we believe that we were open about this already and discussed the various shortcomings, many of which are outside of our control. Nevertheless, for the purpose of our exploratory study, the data quality is sufficient and the benefits of using such a broad dataset outweigh the costs. To assess the consequences of potential errors in the data, we added a sensitivity test in the last version, which was apparently largely overlooked by the reviewer. It appears to us that the

implications of this test might not have been clear enough, and we thus adjusted the description of the corresponding methods and results.

To further explain why we are confident that the sensitivity test supports our view that a few potentially problematic data points will not have a significant impact on the results, we here provide a simplified, less technical overview of the approach and its consequences. As noted in the methods, the input values for the comparative phylogenetic analyses were the median value of the element/Ca ratios per genus. These input values were randomly modified in 100 simulations and then analysed with the same phylogenetic tools. We added a new figure as appendix that displays the simulated values in comparison to the original input value (Fig. A1). For illustration, we list the results of the first three runs below and compare them to the original value per genus:

|  | Original | Simulation 1 | Simulation 2 | Simulation 3 |
| --- | --- | --- | --- | --- |
| *Acrocoelites* | 15.3 | 15.0 | 18.2 | 15.5 |
| *Acroteuthis* | 3.0 | 2.4 | 2.3 | 2.9 |
| *Aulacoteuthis* | 5.4 | 5.6 | 5.3 | 4.8 |
| *Bairstowius* | 9.4 | 9.9 | 11.7 | 9.6 |
| *Belemnellocamax* | 11.6 | 11.0 | 8.2 | 12.0 |
| *Belemnopsis* | 4.4 | 2.1 | 5.9 | 2.2 |
| *Berriasibelus* | 10.6 | 12.1 | 8.1 | 10.2 |
| *Brevibelus* | 13.2 | 13.4 | 14.8 | 14.4 |
| *Castellanibelus* | 10.4 | 9.2 | 10.7 | 11.8 |
| *Conohibolites* | 14.9 | 13.2 | 14.0 | 14.6 |
| *Cylindroteuthis* | 5.1 | 4.9 | 3.9 | 1.8 |
| *Dimitobelus* | 11.3 | 11.3 | 11.8 | 10.3 |
| *Duvalia* | 10.3 | 12.2 | 9.3 | 10.8 |
| *Hibolithes* | 12.7 | 12.0 | 14.2 | 16.0 |
| *Holcobelus* | 3.6 | 2.4 | 3.6 | 1.8 |
| *Lagonibelus* | 5.2 | 4.9 | 6.8 | 4.7 |
| *Mesohibolites* | 16.6 | 17.9 | 15.7 | 17.2 |
| *Nannobelus* | 8.8 | 9.0 | 9.1 | 8.5 |
| *Neohibolites* | 7.6 | 6.2 | 8.6 | 7.3 |
| *Passaloteuthis* | 9.1 | 10.5 | 8.9 | 8.9 |
| *Peratobelus* | 4.8 | 4.7 | 3.3 | 2.4 |
| *Praeoxyteuthis* | 5.1 | 5.2 | 7.5 | 4.7 |
| *Pseudobelus* | 10.9 | 10.3 | 7.7 | 10.7 |
| *Simpsonibelus* | 11.4 | 11.6 | 10.8 | 11.6 |
| *Youngibelus* | 13.3 | 14.5 | 13.3 | 12.1 |

It should thus become clear that the simulated datasets contained values that were significantly different from the original values, more than anything what could reasonably be expected even if diagenetic overprint were more widespread among the data. Note that whether the shift is in positive or negative direction does not matter here, as we cannot know whether the original values are "unaltered" (i.e., a positive shift would simulate the

presence of diagenetic alteration for Mg/Ca) or "altered" (i.e., a negative shift would simulate the "true" values for Mg/Ca). Also keep in mind that this is a modification of the median value for each genus, so depending on sample size, it would actually take a lot of erroneous data to alter the median values by a similar amount. This can be shown by comparing the element/Ca ratios of *Belemnocamax* from the previously discussed Sørensen et al. (2015) study, before and after measurements above the Mn/Ca threshold were excluded:

|  | Mg/Ca | Sr/Ca | Mn/Ca |
|---|---|---|---|
| Original | 11.6 | 2.07 | 0.009 |
| Threshold excluded | 11.5 | 2.12 | 0.005 |

Only Mn/Ca seems to be affected by any reasonable degree. However, this should be no surprise because the threshold was applied to Mn/Ca. In addition, we neither used Mn/Ca nor Fe/Ca in our analyses, so the inclusion of these datapoints has little relevance to our conclusions.

We hope that this demonstrates that the sensitivity test is a powerful tool to assess the uncertainty in the primary data and it shows that our approach is robust against the inclusion of some potentially problematic values. As mentioned by the reviewer himself, no proxy data will be completely free of diagenetic overprint, but since these studies typically target the best-preserved samples, it should not be a major concern for these data – unless the reviewer claims that a majority of previous studies contain significant amounts of problematic data, in which case the consequences would reach far beyond the scope of our paper. The strength of using a probabilistic approach is that we can account for uncertainty, and our results clearly show that uncertainty in tree topology has a more significant effect than uncertainty in the geochemical data for this particular dataset (see Fig. 9). Thus, if anything, we should be concerned about getting an accurate phylogeny before we worry about the geochemical data.

Treatment of diagenetic screening:

The authors are in part reluctant to follow the judgement of authors whose data they utilise in relation to the degree of diagenetic overprint. While this is partially a matter of interpretation, the authors currently include data, which – based on geochemical characteristics – can not be assumed to be unaffected by diagenesis. The statement in line 577-579 in my mind remains problematic in this regard. This point also links back to the quality control of the data used for interpretation, even though the authors seem confident that their approach is robust against minor incorporation of data from poorly preserved samples.

See above for explanations. We are confident that our approach is robust against the inclusion of some diagenetically overprinted data.

I have already made a note with respect to assessing sample preservation in the last review, and this is a point that I am not willing to compromise on. I apologize for a perhaps slightly wordy explanation and drawing largely from my own work to this end, but I feel the philosophy of a geochemical screening approach needs to be outlined here in some detail in case I had not been clear before, and I prefer to use case studies that I am most familiar with. I will here focus entirely on belemnite rostra and omit further complexity arising when studying other fossil calcite:

Employing data from a global survey of published geochemical studies that targeted palaeoenvironmental interpretation of belemnite rostrum geochemistry will lead to a series of biases. Firstly, such studies by and large will be done attempting to sample only the best-preserved material, so the geochemical data arising from such surveys are by no means representative, and therefore skewness and median of such data carry no predictive value. Secondly, such a compilation lumps together specimens derived from a multitude of depositional environments, part of which are characterised by highly specific diagenetic endmembers.

This comment comes as a surprise. Above, the reviewer states that our dataset contains data that is affected by diagenesis, but then he criticises that the data is biased due to the studies "by and large [...] done attempting to sample only the best-preserved material". That seems contradictory to some degree, as it would suggest that there is little diagenetic alteration in the data? Please explain.

The goal of geochemical screening is to make a judgement for each measured sample, whether, based on its geochemical signature, the proxy of interest is sufficiently well preserved to be interpreted from a palaeoenvironmental point of view. Two points are critical to realise here: 1) No belemnite rostrum sample exists that is entirely devoid of geochemical modification subsequent to the biomineralisation process. Some exist for which such modification is negligible, but it is never zero. 2) For this reason, the purpose of geochemical screening for diagenesis is not to identify which sample is overprinted. The purpose is to identify whether a proxy that is target of palaeoenvironmental interpretation has been affected by diagenesis to a degree that it substantially biases the evaluation of this proxy.

Thank you for these comments, as they actually support our approach to the data:

i)    We deliberately excluded Fe/Ca and Mn/Ca, as their distributions already suggest that they are biased and not well suited for the comparative phylogenetic analyses.

ii)    We used sensitivity analyses to identify whether diagenetic overprint would substantially bias our evaluation of Mg/Ca and Sr/Ca ratios through the phylogeny, to which the answer is no.

We agree that palaeoenvironmental interpretations from these data would be problematic, but this is not the point of our study at all.

Typically employed chemical diagenesis proxies for belemnite rostra are Mn/Ca, Fe/Ca, and Sr/Ca. Of these, Mn/Ca is often the least ambiguous, whilst Fe/Ca is typically a little more variable. Sr/Ca (and also the rarely reported Na/Ca) can only act as subsidiary information because the primary Sr content of the rostrum is variable and not exactly known. This complication is even more pronounced for Mg/Ca so that the predictive value of Mg/Ca ratios becomes very limited unless one studies very extreme sedimentary rocks such as dolostones.

Mn/Ca: Compared to the diagenetic endmember, a suitable 1st order assumption is that unmodified calcite of a rostrum has a Mn/Ca ratio of zero. This strongly simplifies considerations of diagenesis, because any measured Mn can then be attributed to incipient diagenetic overprint. The degree of bias imposed by this assumption to my knowledge has never been found to be problematic for belemnite rostra.

The problem is that the primary composition of the belemnite rostrum cannot be known, because belemnites are extinct. It has been repeatedly shown that living biomineralising molluscs may also incorporate certain amounts of Mn into their skeleton. Thus, the assumption that the unmodified ratio is zero is questionable.

Fe/Ca: The same assumption as for Mn/Ca holds, but Fe/Ca data are typically more noisy, perhaps due to the ubiquitous use of metal tools for sample preparation and the vulnerability to lab contamination, as well as perhaps more variable iron sources in sedimentary environments.

Sr/Ca (same holds for Na/Ca): Belemnite rostra overall are typically quite rich in Sr. In most cases (but notably not always), therefore they will lose Sr during diagenetic overprint. This sometimes allows to place a minimum Sr/Ca ratio as a threshold value below which a datum cannot any longer be treated as trustworthy. However, such a threshold can only ever be auxiliary because it is not known how much Sr was lost (due to the natural variability of Sr in rostra); it can only show that the sample in question is definitely problematic when this threshold is crossed.

The question now arises where to set the relevant threshold values. This decision has to be based on a set of parameters: A) What is the diagenetic endmember, and what is its geochemical composition?; B) What is the maximum permissible bias for the target geochemical proxy; C) what is the initial composition of the belemnite rostrum regarding the chosen diagenesis proxy? Note that, if A) cannot be answered because a diagenetic endmember was not established, this screening may well become problematic, sometimes even impossible.

In our opinion, C) is most crucial here, because the original composition cannot be known. There is reasonable evidence that the original composition varied between taxa, and we are by no means the first to suggest this. Thus, if C) cannot be answered, the decision for setting the threshold values becomes arbitrary, as it based on an assumption that is impossible to test. The closest we can get is approximating this value by collecting a large

sample of values from different settings which, according to a range of screening tools, are affected only minimally by diagenetic overprint. This is by no means perfect and likely would require dedicated studies, but it is certainly preferable to merely guessing the original composition.

Case study 1: Belemnite data from Ullmann et al. (2020, Scientific Reports). This study was largely concerned with oxygen and carbon isotope ratios in bivalves and brachiopods, but a few belemnite samples were measured from Toarcian strata in Spain.

For A) It was found that diagenesis shifted the composition of bivalve material towards the composition of bulk carbonate. Whilst sparitic cement was often present, it had distinctive geochemical composition and could be excluded from further consideration as a complicating factor. Median values for diagenesis-relevant geochemical proxies in the bulk rock were: Mn/Ca c. 0.5 mmol/mol; Fe/Ca 19 mmol/mol; Sr/Ca c. 0.4 mmol/mol; Mg/Ca c. 30 mmol/mol; $\delta13C$ c. +2 ‰; $\delta18O$ c. -4 ‰.

For B) The target proxies were carbon and oxygen isotopes, and a maximum permissible bias would have been set around 0.1 ‰ for carbon, and 0.2 ‰ for oxygen isotope ratios (this is the usual 95% confidence interval for analytical data for Mesozoic macrofossils).

For C) The assumption for well-preserved lowermost Toarcian belemnite rostra would be Mn/Ca and Fe/Ca of zero, and about 1.1 to 1.7 mmol/mol for Sr/Ca. Typical $\delta13C$ and $\delta18O$ values in this stratigraphic interval were expected around 0 ‰ and -1 ‰ respectively.

How is the original composition determined? Likely based on previous measurements of countless belemnite rostra! In fact, our Table 4 fits quite well with these numbers, as the central 50% of the Sr/Ca ratio range between 1.4 and 1.6 mmol/mol, though these values vary between genera (e.g., the Early Jurassic *Acrocoelites* displays higher values with a median of 1.9 mmol/mol). Mn/Ca and Fe/Ca ratios are generally low, but rarely exactly zero. Admittedly, there is a problem with values below the detection limit, but since we do not use Mn/Ca or Fe/Ca for our comparative phylogenetic analyses, we consider this point to be of lesser relevance.

From A-C it follows that diagenetic bias on C isotopes would reach 0.1 ‰ for a typical belemnite rostrum after the addition of 5 % bulk-rock like material. This would be associated with a Mn/Ca ratio of 0.025 mmol/mol, and an Fe/Ca ratio of 0.95 mmol/mol. The diagenetic bias on Sr/Ca at this point would be -0.03 to -0.06 mmol/mol depending on original composition.

Diagenetic bias on O isotopes would reach 0.2 ‰ for a typical belemnite rostrum after the addition of 7 % bulk-rock like material. This would be associated with a Mn/Ca ratio of 0.033 mmol/mol, and an Fe/Ca ratio of 1.3 mmol/mol. The diagenetic bias on Sr/Ca at this point would be -0.05 to -0.09 mmol/mol depending on original composition.

Assume that the bulk rock had a 87Sr/86Sr ratio of 0.7100 versus a typical fossil 87Sr/86Sr ratio of 0.7071. Analytical uncertainty of typically c. 0.00002 on this ratio would have been exceeded after the addition of 0.7 % of diagenetic Sr. For example, for a rostrum with original Sr/Ca of 1.5 mmol/mol, this would have been reached after addition of 2.5 % of bulk-rock like carbonate. I.e. geochemical limits of preservation should have been set at 0.013 mmol/mol for Mn/Ca and 0.48 mmol/mol for Fe/Ca. Note that at the stage where 87Sr/86Sr is already significantly affected, the impact on C and O isotopes is still negligible.

In summary, the thresholds set for the proxies of interest are a direct consequence of the diagenetic endmember, the geochemical proxy of interest, and would have been different depending on whether the focus had been on C, O or Sr isotope work.

Thank you for the detailed explanation. What is missing is how much the underlying assumptions of the original composition of the rostrum impacts the thresholds.

Case study 2: UK Toarcian belemnite rostrum from Ullmann et al. (2015).

For A) In this study it was found that geochemical trends of diagenesis were compatible with addition of diagenetic cement, which in strata adjacent to the sampled specimen had Mn/Ca of c. 4.5 mmol/mol, Sr/Ca of c. 1.4 mmol/mol, Mg/Ca of c. 20 mmol/mol, δ13C c. -10 ‰; δ18O c. -15 ‰.

For B) The target proxies were carbon and oxygen isotopes, Sr/Ca and Mg/Ca ratios, and a maximum permissible bias would have been set around 0.1 ‰ for carbon, and 0.15 ‰ for oxygen isotope ratios (this was the 95% confidence interval for analytical data).

For C) The assumption for well-preserved lowermost Toarcian *Passaloteuthis* rostra would be Mn/Ca and Fe/Ca of zero, and about 1.4 to 1.7 mmol/mol for Sr/Ca and 6 to 12 mmol/mol for Mg/Ca. Typical δ13C and δ18O values in this stratigraphic interval were expected around +2 ‰ and -1 ‰ respectively.

Had the same Mn/Ca cutoff been employed as for oxygen isotopes in belemnite rostra from case study 1, the bias would have been 0.1 ‰ on δ18O, and 0.09 ‰ on δ13C. The corresponding Sr/Ca bias would have been -0.0015 to -0.0037 mmol/mol and for Mg/Ca it would have been +0.05 to +0.10 mmol/mol.

Evidently, the only proxy for which such a threshold would have been useful would have been carbon isotopes. Ultimately, a slightly higher Mn/Ca ratio of 0.05 mmol/mol was opted for in this study, because "a Mn/Ca ratio of 0.05 mmol/mol corresponds to the Mn level at which diagenetic effects on oxygen isotopes reach a magnitude equal to analytical reproducibility" (Ullmann et al., 2015).

In summary, relative diagenetic impact on C isotopes would have been stronger in the UK than in Spain. However, due to the much different geochemical composition of the diagenetic endmembers, the effect on O isotopes, Mg/Ca and Sr/Ca would have been subdued for equal Mn addition in UK compared to Spain. On the other hand, adding the same

percentage of diagenetic endmember would have caused much larger geochemical effects in the UK than in Spain, so overall the UK fossils – everything else equal – would have been biased more by diagenesis than in Spain. This closely aligns with my overall observations of fossils from Jurassic strata in these countries.

One could extend this thought experiment further: Diagenetic cement in the Upper Triassic of New Caledonia (Ullmann et al., 2014) have Mn/Ca of c. 2 mmol/mol, Sr/Ca of c. 0.05 mmol/mol, $\delta13C$ c. -20 ‰; $\delta18O$ c. -12 ‰. Clearly, here, carbon isotopes would be very sensitive to very small-scale diagenetic overprints which would leave only a feint signature in Mn. Carboniferous sparry marine cement in Nevada, USA (Brand, 2004) has Mn/Ca of c. 0.2 mmol/mol, Sr/Ca of c. 0.2 mmol/mol, $\delta13C$ c. -4 ‰; $\delta18O$ c. +1 ‰. Again, C isotopes would have likely suffered most notably from overprints, and Sr/Ca to a degree, while even major recrystallisation might not have a big effect on oxygen isotopes. Mn would not have been a very powerful tool to image minor overprints here due to the Mn-depleted nature of these cements. There will be other instances where diagenetic endmembers are rich in Mn, but otherwise geochemically similar to macrofossils, in which case very high Mn/Ca thresholds may be acceptable.

Note that all of the above makes no allowance for any variability in the diagenetic endmembers, and largely also for the studied fossils. Once such variability is encountered it is virtually impossible to avoid false negatives. This is why thresholds for multiple proxies are set so that biased data that are not caught by one proxy by accident can be picked up through another one. Of course this means that a number of samples that may actually be okay will also be discarded for fear of incorporating biased information. Note also, that depending on the used geochemical proxy, additional tools are unavoidable – Mn thresholds are not very good at checking for silicate contamination which is very relevant for B and Li isotope work. Chemical proxies overall may be unable to help with assessing clumped isotope preservation, etc.

I hope the lengthy description above highlights that there is no single geochemical threshold of good preservation. It depends on the studied fossils, the studied site, and the geochemical index of interest. Any attempt to make a global statement about what threshold may be useful is in direct contradiction to these observations and – in my view – misguided. Geochemical thresholds of preservation are imperfect, but – when sensibly employed based on dedicated study of local diagenetic trends – usually (not always) a powerful means of screening for diagenetic bias. For further information I have put together a review for Volumina Jurassica, where some of these considerations feature alongside some recommendations how to sample and study Mesozoic fossils to tackle the issue of diagenesis (Ullmann, 2024, Volumina Jurassica 22, 35-58).

Perhaps we misunderstood the reviewer's comment, but it appears to us that he criticises the use of a single, generally applicable threshold. We never suggested that there is a single "one-size-fits-all" threshold for each element. Instead, we argue that the situation is

more complicated, and even thresholds based on the local diagenetic context can be problematic.

From all of the above, it follows that the definition of threshold limits relies in part on the presumed original composition, for which we show that assuming the same original composition for all belemnite taxa is not correct. It does not matter whether this variability is caused by environmental or vital effects, what matters is the primary composition before or shortly after the death of the animal.

Our view on chemical thresholds is the following: the benchmark for thresholds is the (unknowable) original composition of the biomineral. So in theory, they are independent of the diagenetic setting. With our large collection of published chemical data, we want to approach thresholds from this point of view: getting a statistical approximation of the "original" chemistry of the rostra, based on material that shows as little diagenetic alteration as possible. We made this point clearer in the revised manuscript.

As a compromise, we suggest that when defining thresholds, one should also take the taxonomy into account, and here our comprehensive dataset can provide a baseline, at least at the genus level. Obviously, more research is needed to assess the applicability of taxon-specific thresholds.

Detailed comments:

L14: Here the authors refer to "at least five evolutionary transitions", in L340-1 the authors note "transition from high to low Mg happened at least four times independently". Are these statements compatible?

Thank you for pointing this out, it should be five evolutionary transitions, compare Fig. 7.

L38: The microbial alteration pathway is indeed an option, but not always observed. Especially for organic-rich bands in parts of the rostrum with very little permeability, one may rather expect inorganic processes to dominate, or organic matter to be preserved well (which appears to be the case in a number of instances).

Microbial alteration does not mean the complete consumption of originally present organic matter. It has been demonstrated several times that the organic-rich portions of the belemnite rostra were more permeable. Furthermore, decomposition of organic matter results in new fluid pathways for the subsequent precipitation of calcite cements that occlude these pathways.

L41: I am not sure I follow that "alteration is most pronounced" during burial. This might be true in a number of settings, but there is no reason for this to be a rule. Diagenesis is a

consequence of thermodynamics and reaction kinetics, influenced by biological and physical factors. Its trajectory and pace are set by the unique combination of the above (and changes thereof) over the post-depositional history of the study material.

That seems a misunderstanding, we do not claim that this is the rule, but mention that this contributes to the complexity. The burial realm is characterized by higher temperatures than other diagenetic realms so by definition will in general result in more pronounced diagenetic alteration (as you yourself identify thermodynamics and reaction kinetics as the major factors in diagenesis).

L84: Were only data added into the supplement that had passed such screening? I noted a number of values in this file that had been flagged by the original authors as problematic. Where the authors have a difference in opinion, could such data nevertheless be highlighted?

The supplementary material is much more inclusive than the actual data used (see above). We now highlight data in the supplement that were not part of the analysed data and added an explanation in the data availability statement.

Figure 1: Should step a) not be to "establish a belemnite character matrix"?

The belemnite character matrix was taken from Stevens et al. (2023) and updated to include taxa for which geochemical data is available, which were not included in the previous phylogenetic analysis. Therefore, the "establishment of a belemnite character matrix" was not part of this study.

L90-91: I am not sure I understand how it avoids circular reasoning and maintains comparability if problematic values are not excluded. Besides this, the study by Sørensen et al. (2015) was not limited to element/Ca ratios, but utilised also visual and SEM inspection of the material. This screening supported the interpretation of the addition of diagenetic calcite close to the apical line due to cementation of a somewhat porous central part of the rostrum. Unpublished SEM-based cathodoluminescence images for these specimens exist that show increased luminescence in the areas that were already known to be problematic from SEM secondary electron images and Mn/Ca co-variation with other proxies. One may add that also CL maps for the data in Ullmann et al. (2015) were provided, which support classification of data as overprinted, even though such data are listed in the data supplement for the present work.

For the discussion of diagenetically altered data and thresholds, see earlier comment. The included studies are very heterogeneous, and some applied a stricter screening than others. Thus, it is impossible to avoid at least some diagenetic overprint. Even if the

mentioned studies used a combined screening approach, the sole exclusion criterion seems to have been still the geochemical threshold, and the optical screening was merely used as auxiliary tool. Citing unpublished evidence is hardly a reasonable argument here, does the reviewer expect us to contact every single author to send us all the information that was missing from their publications?

L92: This could be ambiguous for the reader. Is this to say 5 % of the data used in the present work, or 5 % of the data in the quoted study? With respect to the latter, 27 % of the data were excluded due to crossing the Mn threshold.

We change the wording to show that they only minimally change the Mg/Ca and Sr/Ca ratios (see above), for which phylogenetic analyses were performed.

L101: To be consistent with the description in L99, should m(Ca) not be carrying the unit [g/g] and n(Ca) [mol/g]? For consistency this should then be carried over into L104 and L106 as well.

The equation was changed following a request by the reviewer during the first round of reviews. Nevertheless, we adjusted the descriptions and formulas so that it should be consistent now.

L109: It is here stated that intra-generic variability is presumed to be small, but it seems to me that the data shown later do not really support this?

For the purposes of the analysis, the intrageneric variability is comparatively small, with the bulk of the element/Ca ratios falling within a relatively narrow window (Fig. 2). Different species of the same genus appear to be fairly consistent. For *Hibolithes*, if you exclude the extremes such as *H. carparticus*, *H. jaculoides* and *H. obtusirostris*, the median values are quite similar (Fig. 3a). It is similar for *Passaloteuthis*, especially considering that some of the ratios may result from incorrect conversions (Fig. 3c). *Duvalia* appears to be more variable (which may be related to suspected higher amounts of organic matter), but as mentioned in the manuscript, the biggest outlier is *Duvalia* sp., meaning that there is some potential for misidentifications (Fig. 3b) Note also that we are technically talking about the variation between species within a genus, although the term "intrageneric-interspecific" sounds quite cumbersome. We now specify "interspecific variation within the same genus" in the revised version.

Overall, assessing interspecific variation is difficult due to uncertainty in the taxonomic identifications of the material, as these are typically poorly documented and nearly impossible to confirm or correct.

L148: Later, the offset seems to be 216.3 (= middle Norian, Fig. 7 caption). Should this offset value carry a unit (Ma)?

These two offsets are unrelated. Analytically, the former is a specification of the origin prior, while the offset in Fig. 7 corresponds to the root. Note that these two parameters of a phylogenetic tree are not the same, i.e., the origin corresponds to the time when the diversification process started, while the root represents the timing of the last common ancestor of all taxa present in the tree. Moreover, the first offset is a setting on the prior (part of the method), while the latter is from the posterior (part of the result). The prior merely specifies the distribution, from which random values are drawn, and the posterior distribution essentially represents the statistical output values that may be very different from the prior, depending on the model and the data. Technically, different priors lead to roughly the same posterior (unless certain values are specifically excluded by the prior), the main difference being the efficiency of the analysis (i.e., how long they need to be run). The reason why an offset needs to be specified on the origin prior is that logically, the origin cannot be younger than the oldest species in the sample. Also note that the same applies to the root, which is by definition younger than the origin.

Following another request by the reviewer, we changed the figure to show the chronostratigraphy on the x-axis, so the second "offset" is now gone and should not cause confusion anymore.

In comparative phylogenetics, units are typically not reported for the parameter settings. The unit would probably be more confusing than helpful, as this represents a random value that is drawn from an exponential distribution and then added to the offset value. If anything, Myr rather than Ma should be used. However, in this case, lambda would also need a unit, but since the expected mean of an exponential distribution is 1/lambda, the unit of lambda would be 1/Myr. From a pure technical standpoint, no unit is entered into the software, so we think it is not necessary to add a unit directly to the numbers, but we added it in the explanation.

L228: What is the rationale to assume an exponential distribution rather than a normal distribution (and is this symmetric around the average, or an exponential function giving larger emphasis on smaller values)? If data are drawn for which probabilities of being higher is equal to those being lower, then I think this does not adequately reflect natural observations, at least regarding Sr.

See also our earlier comment on the sensitivity analyses. The rationale is explained in L231-232. It essentially reflects a symmetric exponential distribution around the original input value (in the main analyses, we only used the median value of each genus as input). Note that we do not simulate raw data, but a single, altered median value per genus. Most of these values will be relatively close to the original values, but there are some that

significantly differ. Thus, it simulates datasets that would be considered severely biased. The reason for the symmetric distribution is that since we cannot know the "true" element/Ca ratios of each genus, we model both the inclusion of simulated altered data, as well as simulated unaltered data. To take Sr/Ca as an example, if the simulated data are lower than the original data, it would simulate a case where the simulated data are diagenetically altered. In the opposite case, the original data would be diagenetically altered and the simulated data represent the "true" values.

Perhaps this sounds overly complicated, but the message is that we introduced a very heavy noise signal and tested how this affects the estimation of the evolutionary rates. We now include a figure that shows the range of the simulated values (Fig. A2). If these values result in only slight differences in the evolutionary rates, we think that this is a strong argument for the results being robust against the inclusion of some altered data. The uncertainty in the tree topology appears to play a bigger role here.

Table 4: Units are missing and values are reported with much too great precision. I do not think that more than three significant digits are sensible given the analytical uncertainty of such data. I feel values below detection limit should not be excluded from summary statistics. Perhaps an additional column "<LOD" could be added? Detection/Quantification limits should hopefully be listed in the relevant publications.

The explanation under the table seems to lack some words in the second sentence.

We rounded the values to one digit less.

If values are below the detection limit (and thus not reported other than perhaps "<XY"), how can they possibly be included in the calculation of summary statistics? If the threshold limit is taken, it probably represents an overestimation. If these values are assumed to represent zero, it represents an underestimation. As detection limits vary between studies, it is difficult to control for this factor. The supplement lists the detection limit for each measurement, so they can be checked, if necessary. As the detection limit is only relevant for Fe/Ca and Mn/Ca, it is also of minor importance to the study, as we used neither in the comparative phylogenetic analyses.

L244: Would it be possible to quantify this point on variability? I agree that excluding extreme values that are dubious such as explained in L246 is a more realistic way of looking at it.

We added a table to the appendix (Table A1) that shows the variability between and within genera by reporting median, standard deviation and number of measurements. In addition, we provided another figure to the appendix (Fig. A1), where we show the variability of the data, which show that intergeneric variation (expressed as range of median values per genus) is higher than intrageneric variation (expressed as standard variation within each

genus) for Mg/Ca and Sr/Ca, but a similar pattern cannot be ascertained for Fe/Ca and Mn/Ca.

L253: I am not sure that this is a valid statement: These data in my view are not representative for the genera because they are not primary to the biomineral. Their expression is a consequence of diagenesis and the way it was sampled by the quoted studies. My experience is that primary variability of Mn and Fe in rostra is very hard to ascertain, but levels of these metals are (very) low as compared to overprinted samples. This point also links to the statement in L399 – I cannot see any biological reason for the primary incorporation of Mn and Fe into belemnite calcite. What would the sources of these elements in an oxygenated water column be? What evidence is there for this to have occurred to a degree that is meaningfully measurable?

Thresholds: see our earlier reply. Biominerals of mollusks are not formed from seawater but from body fluids (which are modified seawater). Slight amounts of Mn are also present in a modern coleoid cephalopod calcite biomineral: the shells of argonaut octopuses. This is demonstrated by chemical measurements as well as cathodoluminescence (Stevens et al. 2015; 2017). Incorporation of Mn during biomineralization has been demonstrated for several other taxa, including *Nautilus*, by CL-microscopy (e.g., Barbin 2013), even in vertebrates (Hättig et al. 2019). We completely agree that these amounts are much lower than has been assumed in many cases before but it is a demonstrable fact that these elements are primarily incorporated into biominerals.

Furthermore, we do not claim that the data are primary for the genera due to biological control, this is simply data reporting, with the element/Ca ratios grouped after genus. We do not make any interpretation regarding the cause for these distributions but argue that the distributions alone preclude their usability in subsequent analyses. The reviewer argues that the data are not representative of the genera – we effectively argue the same in L253: "[...] the skew may reduce the representativeness of mean or median values [...]". As this belongs to the Results section, we believe that interpretations such as diagenetic effects do not belong here. In fact, we discuss them later in the MS.

Figure 2: Units are missing at axis label (see also figure 3,4,7,11). Would it be possible to add the number of values taken for each genus next to the box plots?

We added the units. We believe that adding the numbers to the plot would make the figure too cluttered (note that some of the box plots span almost the entire width of the figure). However, we added a table to the appendix (Table A1) and refer to it in the figure caption.

Figure 4: Number of values taken for each genus in the individual boxes would be useful to add.

We added the sample sizes for this figure, as it is not included in Table A1 and the spacing of the boxes allows for a better placement of the numbers.

Figure 7: Would it be possible to use Stages and Periods on the x axis rather than a somewhat arbitrary numerical value that is hard to interpret? The use of species names in the labels confuses me slightly. Is this a consequence of the character matrix and the assumption that intra-generic variability is negligible? I thought that Figure 3 spoke against this assumption.

We added the chronostratigraphy as x-axis. Regarding the use of species labels, this is a consequence of the use of species for the morphology-based phylogenetic analysis from the previous step. It is true that the element/Ca ratios reflect the genus level instead, so we adjusted the labels accordingly. This also allows for some more space in the plot. The labels were also adjusted in Fig. 8., which allowed for a better arrangement of the subfigures.

Figure 7&8: The match is not perfect, but besides phylogenetic links, there seems to be a temporal pattern of changes in Mg and Sr with for example an aggregation of taxa with low Mg/Ca and low Sr/Ca approximately in the Late Jurassic-Early Cretaceous. Given that seawater Mg/Ca and Sr/Ca are thought to have fluctuated by considerable amounts, could there not be a signal of ocean chemistry embedded into this plot?

L404: Why should there not be a trajectory in Sr/Ca and Mg/Ca that leads to absolute changes in these ratios (see point above for fig. 7&8)?

This is simply a description of the changes or, more precisely, the rate of these changes, without implying causality. This is already stated in L409: "We also stress that $\sigma^2$ is simply a metric for the tempo of changes in the trait value, independently of the underlying causes."

L425: Perhaps I misunderstand, but do the authors mean "systematic" here instead of "non-systematic"?

No, because systematic shifts mean that all ratios are modified by a constant factor. In the discussed case, all belemnite rostra would record the same shift, independent of their taxonomy. Thus, under the assumption that proxy data only reflect seawater properties, investigating these changes across the phylogeny can inform how clades responded to climatic changes (e.g., do clades remain within the same temperature range, record temperature shifts or go extinct?). Our point is that even if seawater were the only relevant factor, the investigation of evolutionary rates of geochemistry can provide important insights into the influence of climate on the evolution of belemnites.

L424: There is no reason to assume that diagenesis causes homogenisation of data. Incomplete diagenesis in a local context will produce increased heterogeneity of the data (this is what enables us to establish geochemical trajectories of alteration in the absence of a known diagenetic endmember). Complete overprint would lead to relatively uniform data within a (short) succession, but the absolute values would then be set by the local diagenetic endmember. On a global scale, a combination of completely overprinted data would be anything but uniform.

Yes, but it would be expected that the differences between genera would disappear due to this increased heterogeneity. Thus, while the distributions of element/Ca ratios would get wider as a whole, the differences between genera would disappear, because it depends on the diagenetic endmember and not the taxonomic identity. We adjusted the statement to reflect this.

L450: See comments to Fig. 3 and L109.

See our replies above.

L460: I had already pointed this out earlier, but there is currently no study that can support unequivocally that there would have been a significant decrease of Mg/Ca and Sr/Ca through ontogeny in belemnite rostra. Data by Stevens et al., 2017 only show one half-profile per specimen so do not allow to disentangle crystallographic, rate, and age effects on Mg and Sr incorporation. Stevens et al. (2022) report three half-transects through one Duvalia rostrum, but the number of interpretable data is so low that a coherent image of element incorporation controls can not be constructed. Data from Ullmann et al. (2015) relate to four complete profiles through a single rostrum. These data show highly significant increases in Mg/Ca and Sr/Ca towards the apical line that could be understood as high Mg/Ca and Sr/Ca in early ontogeny. However, considering their figure 10, it is noteworthy that Mg and Sr content in material formed close to the apical line in profiles progressively closer to the apex of the rostrum is much higher than time-equivalent calcite closer to the protoconch. Furthermore, Mg and Sr enrichment in calcite close to the apical line near the apex is greater than Sr and Mg enrichment towards the apical line near the protoconch. If anything, these data indicate that calcite unaffected by crystal shape that might have formed during early ontogeny would have been similar or somewhat depleted in Mg and Sr compared to average values in the specimen.

The principal control on the formation of Mg- and Sr-rich calcite in the rostra should therefore be the closeness to the apical line, not the age of the belemnite.

One could argue here that the study of Ullmann et al 2015 is also not representative of belemnites as a whole, sampling only one specimen. Stevens et al. 2017 also showed

that a distinct ontogenetic stage (the epirostrum) had different chemical composition than the orthorostrum. In the end this is semantics: a rostrum also grows lengthwise, meaning that the data of Ullmann et al 2015 are indeed the Mg/Ca and Sr/Ca data fluctuating during ontogeny. We do not dispute that closeness to the apical line has no effect. We adapted this section accordingly.

L477: This is a slightly too pessimistic outlook in my mind. If significant changes in composition through time can be established in taxa that occur over long periods of time, and/or numerous taxa without close phylogenetic links, linking such changes to ocean chemical composition would be a reasonably hypothesis. Admittedly, there are very few if any periods of geological history for which this can be assessed without some doubt at present due to the lack of data coverage.

We adjusted the statement but maintain that calculating these coefficients would be challenging.

L484: I do not think that sampling bias would cancel out in the grand scheme of things. Discounting complications of diagenetic overprints, an indiscriminate sampling of calcite in the belemnite rostrum should create a dataset with a skew towards high Mg/Ca and high Sr/Ca. There is no sampling approach that would create an opposing effect. Consequently, one either gets datasets with an enrichment in high Mg and high Sr values, or one that does not. The bias is unidirectional.

Figure 2 does not really support this, as outliers exist in either direction, though it depends on the proxy of interest. Nevertheless, the important thing is that sampling or analytical bias would be indiscriminate of taxonomy, meaning that even if the bias is unidirectional, relative differences between genera would remain, although perhaps with higher variability within each genus. We rephrased our statement to clarify this point.

L506-10: I still do not see how these considerations are meaningful for belemnite rostra: as stated also by the authors, all belemnite rostra considered here are thought to be primary calcite. Hence, the same crystallographic Mg and Sr incorporation control applies to all of them. This precludes a preference for Mg and Sr in some of them based on carbonate polymorph. For sure, more Mg can be incorporated into calcite than into aragonite and so there is a wider range of possible expression of Mg/Ca in calcite than aragonite. For Sr however, primary Sr content in biological low-Mg calcite can be as low as c. 0.5 mmol/mol in some modern brachiopods and > 5 mmol/mol in some barnacles. Some modern aragonite-secreting taxa have lower Sr/Ca ratios than belemnite rostra (see Dodd, 1967, Journal of Paleontology 41, 1313-1329 for an early overview), speaking against a dominantly carbonate-polymorph control on Sr content.

R: Aragonite does not really matter here, only the Mg vs. Sr incorporation in calcite is relevant. Our point is that, as explained by the reviewer, there is a wider range of possible expressions of Mg/Ca in comparison with Sr/Ca. This corresponds to the pattern we see in Mg/Ca, which more strongly aligns with the phylogeny than Sr/Ca, where the distribution is more homogeneous (see Fig. 7, 8). We slightly reworded our statement.

L573: Is not this statement and the one in L578 in contradiction to L579?

See earlier comment.

L591: I do not think this is necessarily crucial. There are numerous studies that benefit from taxonomic identification, but for some it is irrelevant. For instance, for reconstruction of the marine 87Sr/86Sr curve it is not a requirement, because biological fractionation of Sr isotope ratios is adequately corrected for through normalisation to a common 84Sr/86Sr ratio.

For 87Sr/86Sr this is indeed a valid point but this is about the only case in which a taxonomic identification can be considered irrelevant (at least from a geochemist's point of view). One of the major problems with the lack of taxonomic assignment is that newly derived and published data might be of use in other studies for which taxonomy might play a crucial role (as in the present study). If the primary data lacks any taxonomic information, it might be unusable by other researchers in the future.

L599: The debate about chemical parameters in the rostrum within this study promotes the idea that this variability is purely a consequence of biological controls. I do not think that this is viable given the global tectonic changes and probably changes in seawater dissolved metals through time, but at least one may reasonably expect that there is a dominant biological control on incorporation of these metals. Once C and O isotopes are targeted, this condition does not hold any longer. It may be known that some belemnite taxa had distinct vital effects at least on C isotope incorporation, but these were limited to a few permil even in relatively prominent cases. Over the course of the existence of belemnites, average C isotope composition of the exogenic carbon changed by much more than that, and seawater – besides planetary changes, was almost certainly strongly heterogeneous in terms of latitudinal and bathymetric O isotope ratios and temperature. It would be very challenging then to extract out of such a complex system any data with phylogenetic meaning. Unless species-specific offsets from seawater equilibrium could be obtained (this would be very challenging if at all possible with sufficient precision), the isotopic signatures of taxa would simply represent palaeotemperature and state of the carbon cycle at the time of their existence.

The conclusion that our study "promotes the idea that this variability is purely a consequence of biological controls" comes as a surprise. This is a fundamental misunderstanding. We never state this and extensively discuss different factors (e.g., L440-485 in the previous version). In the abstract, we state: "This study highlights the complex interplay between evolutionary, ontogenetic, environmental and diagenetic effects". We furthermore explain why investigating evolutionary rates is meaningful even if purely environmentally controlled. If anything, it makes the interpretation more straightforward because the biological factor is negligible. Obviously, the goal would not be to use them as a character for phylogenetic analyses. Imagine a phylogenetic tree, for which each species has associated temperature values – this would be incredibly useful for evolutionary biologists, as it allows to investigate the impact of local climatic changes on the evolution of a clade. Currently, the best we can do is put a global curve behind a phylogeny and try to interpret how this affects evolutionary patterns. Thus, we believe that investigating C and O isotopes have perhaps even more potential than element/Ca ratios.

We would also like to remark that in the last round of reviews, the reviewer specifically asked why we did not use C and O isotope data, as they would have been readily available.

Thank you for your professional work!

Alexander Pohle, on behalf of the authors

---

## Author Response (AR3)

Dear Editor Niels de Winter,

Thank you for the final round of reviews. We implemented all the remaining suggestions from the reviewers according to the responses below (in red). We are looking forward to seeing our article published and thank you for your professional work.

**Editor:**

- Consider deleting or rephrasing the statement on line 85 about the use of data flagged by the original authors as altered if the procedure for excluding these problematic datapoints is not strictly followed in the dataset.

We rephrased this statement.

- Please clarify in the text whether median values and associated standard deviations or errors were computed by weighting data per species or in an unweighted manner (averaging all datapoints within a genus).

No weighting was used. As this was never implied anywhere in the previous version, we think that this should be clear already, but we nevertheless added a statement.

- Consider rephrasing the statement on line 614 in such a way that it is clear whether or not fixed thresholds for trace element chemistry would be appropriate as a way to screen for diagenesis. I feel that there is broad agreement between the reviewer and authors (and myself) that there is no single "one-size-fits-all" threshold for preservation and that the trace element concentration in a fossil (belemnite) does not solely depend on the "degree of alteration" but on circumstantial parameters such as the geological context and variations in the original in vivo carbonate composition for the taxon.

We rephrased this statement.

-Please clarify your recommendation for using stable isotope data in a phylogeochemical study. Perhaps adding a short statement highlighting that one should take into account long-term and spatial variability in the environment in such an approach would be helpful.

Yes, this is a good point, we clarified the statement.

- Consider clarifying in the discussion whether evolutionary trends can be confidently isolated from effects of diagenesis and environmental change on the trace element data. If this is not possible, I agree with the suggestion by the reviewer that "evolutionary" be dropped from the statements in the Abstract ( line 10 and 14).

As outlined in the manuscript, it is challenging to separate these factors. Nevertheless, we believe that using only "rate" is somewhat unspecific, as it could also refer to a purely temporal perspective. The strength of our approach is specifically that we take evolutionary relationships into account. Therefore, in our opinion, it still makes sense to use "evolutionary", even if biological control is not the only factor. Furthermore, the term is established in evolutionary biology, where it is calculated using the same or similar methods. Thus, it would probably cause more confusion than would be solved by dropping "evolutionary". As a compromise, we replaced "evolutionary rate" with "phenotypic evolutionary rate" throughout the manuscript, as the phenotype may be influenced by both, genetic and environmental factors (in contrast to the genotype). For Mg/Ca and Sr/Ca, which are the only ones for which phenotypic evolutionary rates were calculated, diagenesis plays likely only a minimal role, as outlined before.

**Reviewer:**

Following the second round of reviews for the study "Phylogeochemistry: exploring evolutionary constraints on belemnite rostrum element composition", the authors have comprehensively responded to remaining comments and made further modifications, which I think have been very helpful to clarify the message of this contribution.

Thank you for your feedback!

Differences in opinion remain, but largely I feel that the last round of reviews and the extensive responses of the authors have been helpful in highlighting how certain points they were hoping to make were interpreted differently by me. At this stage it is clear that any differences in opinion will not be resolvable, which is not a problem in any case. I think the focus should now be on resolving residual potential for misunderstandings of what the authors stated they wanted to convey.

Thank you for your understanding. Even if different opinions remain, we believe that this is important to move science forward and thank the reviewer for the critical comments.

Use of data that have been flagged as problematic by previous authors

The authors have been clear and thorough in demonstrating that the inclusion – even of substantial numbers – of problematic data into their analysis has very limited impact on their interpretation due to the use of median values. I do not contest this to be true. I

nevertheless felt that the inclusion of data that have been shown to be unrepresentative of a taxon beyond reasonable doubt would be unfortunate. Readers may get the impression that these data are in fact valid. The authors note in L85 that they did not include data where evidence for alteration had been found, yet then state that they do so in certain cases (L 91). This approach seems unnecessarily arbitrary to me and – in my mind – does not really reduce circular reasoning, nor does it make data more comparable between studies (L90). For the benefit of the reader, I feel that either the statement in L85 ought to be deleted, or be followed stringently.

See comment above to editor.

Use of data that are (likely) erroneous

It is helpful that the authors included now in L114 that there is definite potential for some of the used Mg/Ca ratios to be incorrectly reported in the original paper and outline in L277 onwards what the consequences for this would be. From my point of view, a number of points speak for the listed ratios in the supplement to be computed mistakenly: The data were published by the same lead author in close temporal proximity, the same type of fossils and region were studied, there is the occurrence of a simple conversion error relating to Sr, the supplementary data files for both studies look nearly identical in design, and if one was to follow this interpretation, the Mg offsets that are otherwise observed would largely disappear.

However, a remote possibility remains that the data are in fact valid, and that has implications for other interpretations: If the published Mg/Ca data are correct and indeed low, what is the reason for belemnites for the studied region to be chemically distinct? Following on from this, if intra-generic differences can be so big, how much weight can be given to perceived evolutionary changes in Mg/Ca at higher level, especially where based on few measurements of only a small number of rostra?

The authors added some text towards this matter in L266 onwards, but I am not sure how methodologically this was computed and how robust this statement is. Were intrageneric s.d. values derived from s.d. of all individual measurements for this genus, i.e. unweighted, or for the averages (median) of individual species studied within the genus, i.e. weighted. My guess is the former, having tested this using data on Passaloteuthis: I have replicated the dataset as plotted in figure 3c and found that the median Mg/Ca for these species is 4.0 mmol/mol with 2 s.d. of 5.9 mmol/mol when using only the median values for the individual species and for Passaloteuthis sp. (n = 10). When using the entire dataset for Passalotheuthis, which is dominated by P. sp. and P. bisulcata, the result instead is a median of 9.1 mmol/mol with 2 s.d. of 4.4 mmol/mol (n = 569). If opting to show standard deviations, I think it would be insufficient to only use 1 s.d. values for comparison – 2 s.d. at least give 95 % confidence in the range. However, I am not sure whether it would be more appropriate to use the standard error

of the mean here, if the authors want to justify the robustness of the derived median values? However, as evident from the above, given the heterogeneity of the dataset it would drastically change the median value at least for this genus depending on how it was computed – much more so than what the standard error of the mean would suggest. Based on the entire dataset, a robust average of any sample taken from any Passaloteuthis would be expected to be 9.1 +/- 0.2 mmol/mol (2 s.e., n = 569), while the expected median for any species of Passaloteuthis would be 4 +/- 2 mmol/mol (2 s.e. n = 10).

Regarding the issues which are most prominent with Passaloteuthis, but impact also on other taxa, I have contacted the lead author on the affected studies to ask for clarification. Unfortunately, I have been unable to clarify the matter with her entirely before the review was due, but I do hope the authors can do so and will follow up on this point with her.

See comment above to editor. Note that species identification had no influence on the calculation of our median values and we never calculated any summary statistics at the species level. This would probably make not much sense, as (i) many species are represented only by a single sample; (ii) there is no way to verify the identifications; (iii) there are many taxa in open nomenclature (e.g., *Passaloteuthis* sp.), meaning that they may contain multiple species or are, in part, identical with other species in the dataset.

Regarding intra-generic differences between species: The taxonomy of belemnites can in no way be considered "final". From a modern biological standpoint, only monophyletic groups should be accepted, but this has never been tested, and phylogenetic approaches have just been started by our group. Genera and to a certain extent even species are categories defined by humans and not "natural units". There are many species today that can hardly be distinguished by morphology alone, sometimes this even extends to the genus level. Thus, it is very well possible that some of the species in the dataset should actually belong to a different genus. If geochemical signatures can be corroborated to be characteristic for certain taxa and correlate with other traits (e.g., morphometric data), this could someday maybe be used in taxonomic diagnoses. As of now, this is again a simplification but is currently the best we have. It highlights again that dedicated studies focusing on individual species with very restricted temporal and spatial range is the way forward. Such a study would be independent of the genus, as in the end, this is just a label.

Statement on diagenetic screening:

I am more comfortable with the way in which the text in section 4.3 has been updated and from the responses to the earlier review I sense that the authors are thinking largely along the same lines as I do. Just for the benefit of avoiding any confusion amongst the readers then I think a few points could benefit from final tweaks:

L608 notes that earlier studies often applied too high limiting element concentrations, which they tentatively and partially (L608-9) link to the lack of knowledge about pristine composition of fossil. They further caution in L612-13 that taxonomy and local diagenetic context need to be taken into consideration for confident assessment.

I fully concur that taxonomy and local diagenetic context are crucial to this, and it may also be true that previous authors have been a little optimistic when it comes to their screening thresholds. Where I think the disagreement lies is whether or not the original composition of the belemnite rostra is unknown, or at least what the extent of "unknown" is. In my opinion, to a practical limit it actually is known relatively well for a number of geochemical proxies and taxa, but I sense that the authors feel otherwise.

The example that the authors chose is found in L95: Given current knowledge we can assume that a representative Mg/Ca ratio of B. mammillatus is c. 12 mmol/mol, and for Sr/Ca the ratio is around 2.1 mmol/mol regardless of whether one thinks any of the published data are overprinted or not. To me this means that for these two element/Ca ratios, we have good knowledge of original composition. We can even constrain the likely data spread to allow future studies to compare against: Mg/Ca of 9.6-13.3 mmol/mol for 90% quantile and Sr/Ca of 1.9-2.4 mmol/mol for 90% quantile of screened data.

The authors rightly caution that we do have no proof that original composition for Mn/Ca and Fe/Ca is zero, an assumption that is implicitly made when assigning Mn and Fe thresholds. The authors (if I understand correctly) feel that this invalidates the use of these ratios. While ultimately I fully agree that we have no exact knowledge of the primary range of Mn and Fe values in any taxon, we tend to have quite reasonable constraints on this regardless. The question is whether this meaningfully impacts on their use for screening purposes, i.e. if it makes any practical difference if original Mn/Ca and Fe/Ca were zero, or some small finite quantity.

Following on with the same example that the authors note in the paper, we can only discuss Mn/Ca, but Fe/Ca would follow analogous reasoning. The observation is that the median Mn/Ca ratio in the tested materials was found to be 9 µmol/mol, so following above arguments for Mg and Sr this would be a reasonable maximum estimate for primary median Mn. For the studied specimens – distinct geochemical co-variation with Sr, C isotopes and Mg/Ca arises at Mn/Ca levels > 20 µmol/mol. C isotope values heavier than any other observed values and Sr/Ca ratios lower than any other values are seen in the samples from 20 µmol/mol upward in two out of three specimens. From this it follows that 20 µmol/mol signify (for this basin, time and taxon), a level where alteration is prominently developed, even though it cannot be excluded that some samples may inadvertently have been excluded that originally had Mn/Ca ratios > 20 µmol/mol. It can also not be excluded that samples with Mn/Ca ratios lower than 20 µmol/mol were retained despite changes to their geochemistry, but the geochemical proxy data for these samples do not markedly differ from seemingly well-preserved

material. It is important also to add, that samples that are excluded on the grounds of Mn/Ca ratios exceeding Mn/Ca ratios of 20 μmol/mol are nearly invariably found either in coherent sequences including the apical line or the rim of the rostra, which are areas that are well known to be most affected by diagenesis. It would thus be an odd circumstance for primary incorporation of Mn being most pronounced in these areas which are otherwise known to suffer from preservation issues and co-occur with values in other geochemical proxies that are most removed from the typical values for the taxon. The apical zone of the studied taxon has been shown to still be partially porous using petrographic techniques, increasing the likelihood of encountering diagenetic cements here, and geochemical trends prescribed to alteration point towards values akin to Late Cretaceous chalk (as known from other studies).

Given the above, does it practically matter that primary Mn/Ca ratios are not exactly known? Certainly not for any of the studied proxies whose distribution is not meaningfully affected by any choice of Mn limit lower than 20 μmol/mol. Mn itself is not interpreted and this will probably always be the case given the ubiquitous minor enrichments of this element even in samples that have seen very little diagenetic overprint. I would also add that, while it ultimately boils down to just one number (20 μmol/mol), this number was chosen based on varied evidence from multiple screening techniques, respecting the taxonomic and regional geological context. This is I think in keeping with what the authors argue should be done as well and includes (but goes beyond) "getting a statistical approximation of the "original" chemistry of the rostra". The same approach would have been taken in numerous other studies that opted for other, locally appropriate limiting values based on varied considerations and a multi-proxy approach.

The authors note in their response to the last review that they "never suggested that there is a single "one-size-fits-all" threshold for each element". As a reader I would take the statement in L614 differently. This line indicates that a single value may "give hints for more refined thresholds", even though the cited table does not consider taxonomy, nor local geological context. I would just like there to be no opportunity to misunderstand the approach that I think both authors and myself feel is more appropriate.

Thank you for your detailed explanation. We implemented the change suggested by the editor (see above), so this potential for misunderstanding should now be avoided. At a very broad level, Table 4 would provide information on the order level (Belemnitida), but of course, it is preferable to consider at least the genus-level.

The use of C and O isotopes for comparable studies

The authors emphatically note in their response to the last review that C and O isotope ratios would not be used in the same way as median Sr/Ca and Mg/Ca ratios. My remark

in the previous review was to ask specifically, why such data were not utilised if they are available and useful for this kind of analysis, driven by the authors' proposal in the conclusions to do so, and by my attempts to understand how these data would be usefully applied. I agree that knowing temperature values for individual species would be great, but this is not something that can be readily obtained for belemnite species using a global median of O isotope data for each species. Things become even trickier for C isotopes. For instance, Passaloteuthis is known from the Pliensbachian and lower Toarcian from European sections as well as Russia; this period saw large-scale palaeoclimate and carbon cycle shifts. Which – if any – C and O isotope value out of the range recorded by the genus throughout its existence as a consequence of changes and heterogeneity in carbon cycle, palaeotemperature, and other environmental parameters, would meaningfully describe this genus?

I cannot help but read L637 which calls for "comparable studies for isotope data" in a way that C and O isotope data should be used in analogy to Sr/Ca and Mg/Ca. As a reader I would assume that this means one should use median C and O isotope data to add them into the mix of proxies that were studied in the present contribution. If this is not the intention, then perhaps just rephrasing this statement a bit would be useful?

Thank you for this point. Of course, there will always be necessary adjustments in the study design. The reviewer rightly points out that global averages of genera would not be appropriate. Nevertheless, this can be solved by only sampling specimens (ideally only a single species) from the same locality and a very restricted stratigraphic range). We clarified this in the revised discussion.

Clarity of message

I am now not fully sure what ultimately the authors' assessment of the main underlying cause of the observed patterns in their data is given that they contest my interpretation (in relation to L599 of previous review) that they concluded the data to largely reflect biological controls.

It is correct that the authors repeatedly state that interpretation of Mg/Ca and Sr/Ca data is complex, for example prominently so in the abstract (L10) and point this out also in their last set of responses: "This study highlights the complex interplay between evolutionary, ontogenetic, environmental and diagenetic effects".

However, they equally state that signatures are dominantly taxon-specific (L10), and display high evolutionary rates (L11), with "five evolutionary transitions" (L14). Within the text it feels to me that the authors ultimately build towards section 4.2 which promotes the idea that taxon-specific element uptake is the principle feature targeted by their dataset, and the statements in lines L545-547 introduce this by relativising any of the ontogenetic, environmental and diagenetic effects as things that can be

circumvented by specific sampling routines. Diagenesis had also already been discounted earlier in the study as a relevant factor (L95).

As a reader I would assume that "evolutionary rates" and "evolutionary transitions" are controlled by the taxa, and not by environmental change. Otherwise they would just be "rates" and "transitions"? I feel this is more than a semantic issue. If the element/Ca data are changing through time because of external factors, they do not perhaps mean very much in terms of evolutionary lineages of belemnites. If instead they are an expression of genetic changes in belemnites, and display large, repeated swings, this would lead to much different conclusions. If the authors feel unsure about the underlying causes of these changes, then perhaps "evolutionary" should be dropped or replaced for clarity?

See comment above to editor. Note that "taxon-specific" does not imply a genetic mechanism – different taxa live at different times in different habitats/environments, which could be another reason for these patterns. We replaced "evolutionary transitions" from the abstract.

Figure 7: I appreciate the extra work adding a chronostratigraphic chart as a new x-axis here. I think that this really improves readability of the graphs, even though perhaps the Cenozoic part of the plots is not required. It seems, however, that accidentally the wrong TJ-boundary age was added here.

Thank you for pointing this out, the age of the Lower/Upper Jurassic boundary was accidentally used in the previous version, which is now corrected. We agree that the Cenozoic part is perhaps not required, but since the names of some of the belemnite genera are relatively long, it avoids having excessive amounts of white space.